# Community structure unveils the path multiplicity in complex networks

Ye Deng[1,2], Jun Wu [1,2] ✉, Xin Lu [3] ✉, Petter Holme [4,5], Daqing Li [6,7] ✉, Zengru Di [1,2,8], Guanrong Chen [9] & Jürgen Kurths [10,11]

Networks with complex topologies describe numerous natural and social systems. Recent studies on path multiplicity have shown strong heterogeneity in shortest paths between node pairs in real-world networks. However, the mechanism underlying this phenomenon remains unexplored. Here, we reveal that community structure is a key factor shaping path multiplicity. To explore the intrinsic factors that influence path multiplicity, we first introduce the concept of relative path multiplicity and find that community structure is more strongly correlated with path multiplicity than other network metrics. Through targeted edge-rewiring experiments, we verify the link between path multiplicity and community structure. The underlying mechanism can be interpreted as an interface-driven effect that sharply increases the number of shortest paths. Inspired by these findings, we propose a tribal-structure-based network model that reproduces phenomena observed in real-world networks. Our work enhances the understanding of network organization, with potential applications in network design and optimization.

Complex networks form the backbone of many natural and technological systems and are crucial to revealing structural and functional relationships in a wide variety of domains[1,2]. From social interactions and biological processes to technological systems, these networks are characterized by complex connections among their components[3–6]. Despite the diversity of their applications, complex networks often exhibit common topological properties, including small-world effect[7,8] and scale-free structures[9,10], which have become central themes in network science[11–25]. In the study of complex networks, paths between node pairs play a critical role in understanding the dynamics and efficiency of the network. Traditional research has largely focused on the shortest routes between nodes, which serve as a fundamental measure of network connectivity and routing efficiency[2]. The small-world phenomenon, in which even large networks often have surprisingly short path lengths, has been extensively investigated[7,26]. Beyond the length of the shortest path, the centrality of the interconnections has been widely studied as a measure of the influence of nodes (edges) on the basis of their frequency of appearance on the shortest paths between other node pairs[27,28]. This metric is thus essential for analyzing network flow and information diffusion and identifying key nodes[29]. Furthermore, research on navigability, namely, how easily nodes can be reached through paths, has been conducted to explore how decentralized routing decisions are made in networks[26].

Despite the advances in the study of paths, path multiplicity—the number of equidistant shortest paths between node pairs—remains unexplored. Recently, some studies have revealed the existence of a universal power-law scaling of path multiplicity, indicating that a few

[1]Department of Systems Science, Faculty of Arts and Sciences, Beijing Normal University, Zhuhai, China. [2]International Academic Center of Complex Systems, Beijing Normal University, Zhuhai, China. [3]College of Systems Engineering, National University of Defense Technology, Changsha, China. [4]Department of Computer Science, Aalto University, Espoo, Finland. [5]Center for Computational Social Science, Kobe University, Kobe, Japan. [6]School of Reliability and Systems Engineering, Beihang University, Beijing, China. [7]Department of Science and Technology, Civil Aviation University of China, Tianjin, China. [8]School of Systems Science, Beijing Normal University, Beijing, China. [9]Department of Electrical Engineering, City University of Hong Kong, Hong Kong SAR, China. [10]Research Domain Complexity Science, Potsdam Institute for Climate Impact Research, Potsdam, Germany. [11]Department of Physics, Humboldt-Universität zu Berlin, Berlin, Germany. ✉e-mail: junwu@bnu.edu.cn; xin.lu.lab@outlook.com; daqingl@buaa.edu.cn

node pairs have a large number of shortest paths[30]. For example, even in the Bn-Macaque-Rhesus-Brain-1 brain network with only 242 nodes[31], the largest count of shortest paths between node pairs can reach 649, with an average value of 11.07. This observation reveals that the world we live in is not just a "small-world", but also a "hesitant-world". The "small-world" effect means that most nodes are connected through only a few steps, even in a large network; meanwhile, the "hesitant-world" effect implies that one may hesitate among numerous choices, even when the network itself is small.

Although the multiplicity of shortest paths is a structural feature, it actively governs how networks function. First, path multiplicity shapes a network's robustness and vulnerability. While redundant shortest paths can preserve connectivity when links fail, they may also concentrate traffic on common bridges or hubs, creating bottlenecks that increase susceptibility to congestion or targeted attacks—a dual role noted in studies of network flows[32]. Second, it affects spreading processes and transport. Multiple, equally short routes can accelerate the propagation of information, diseases, or resources by offering parallel transmission channels, while also redistributing loads across the network and influencing diffusion outcomes[20]. Finally, it has consequences for routing and decision-making. For any agent—whether a packet, an individual, or an algorithm—navigating among many "optimal" paths can induce "choice overload," a psychological effect known to impair decision efficiency[33], potentially degrading the performance of navigation strategies[26].

Our recent finding[30]—a strong power-law distribution in path multiplicity across diverse real-world networks—reveals that the path multiplicity is a fundamental and widespread architectural signature. Given the functional relevance outlined above, a central unanswered question emerges: which underlying network structure produces such strong heterogeneity in path multiplicity? Uncovering this structural origin is essential for explaining observed network dynamics and designing networks with desired functional properties. In this study, we introduce the concept of relative path multiplicity to systematically compare real-world networks with equivalent random networks. Using this approach, we identify the structural factors that shape path multiplicity in complex networks. Furthermore, we develop a network model that can reproduce the empirical characteristics, offering mechanistic insights into the observed power-law distribution of path multiplicity.

## Results

### Concept of relative path multiplicity

Let $G(V, E)$ be a simple undirected graph representing a complex network, where $V$ is the set of nodes and $E \subseteq V \times V$ is the set of edges. Let $N = |V|$ and $M = |E|$ represent the numbers of nodes and edges, respectively. The adjacency matrix of the network is denoted by $A(G) = (a_{ij})_{N \times N}$, where $a_{ij} = a_{ji} = 1$ if nodes $v_i$ and $v_j$ are connected and $a_{ij} = a_{ji} = 0$ otherwise. The edge density of the network is defined as follows:

$$p = \langle k \rangle / (N - 1), \tag{1}$$

where $\langle k \rangle$ denotes the average node degree.

For any pair of nodes $(v_i, v_j)$, the shortest path length between nodes $v_i$ and $v_j$ is denoted by $l_{ij}$. Then we define the *path multiplicity amount* (PMA) between two nodes as the number of shortest paths $h_{ij}$ connecting them, and the matrix $H(G) = (h_{ij})_{N \times N}$ is called the *path multiplicity matrix* (PMM). To quantify the overall multiplicity of the path of a network, we define the *path multiplicity index* (PMI) as the average value of the PMA in all pairs of nodes. Mathematically, this relationship is expressed as follows:

$$\Phi(G) = \frac{\sum_{i=1}^{N} \sum_{j \neq i} h_{ij}}{N(N - 1)}. \tag{2}$$

The PMI reflects the complexity of path selection within the network: a large PMI suggests that the network has a hesitant-world property, with many shortest paths between node pairs, potentially leading to a more complex decision-making process. To reveal the relationships between the PMI and classical network metrics, we present corresponding scatter plots along with the Pearson's correlation coefficient $\rho_P$[34], Spearman's correlation coefficient $\rho_s$[35], and the quadrant count ratio (QCR)[36,37] in 140 real-world networks in Fig. 1a. Pearson's correlation coefficient quantifies the strength of a linear relationship between two variables, whereas Spearman's correlation coefficient is a nonparametric rank-based measure of monotonic association and is often preferred when nonlinearity or outliers make Pearson's coefficient undesirable. The QCR provides a coarse-grained measure of association by dichotomizing each variable at a central location (typically the median) and quantifying how often paired observations fall in the same corresponding half (see the "Methods" section for details). Together, Pearson's correlation, Spearman's correlation, and QCR offer complementary perspectives on association by targeting linear association, monotonic rank-based association, and median-split concordance, respectively. Consistent results across these measures provide a robustness check for the reported association. Overall, the scatter plots exhibit a disordered distribution, with points broadly dispersed, showing no clear trend. Moreover, it can be observed that all three correlation metrics are relatively low, indicating that there is no significant correlation between PMI and classical network metrics.

Intuitively, according to the definition of PMI, edge density may affect path multiplicity. The simple example network shown in Fig. 1b is initially a chain network with an initial PMI value of 1. As the edge density increases, and thus the network becomes fully connected, the corresponding PMI value returns to 1. Furthermore, the changing trend of PMI as the edge density $p$ increases is shown in Fig. 1c for three typical model networks: Erdős-Rényi random networks (ER)[38], regular ring lattices (RRL)[39], and cluster Barabási-Albert scale-free networks (CBA)[40]. Unsurprisingly, one can see that the PMI values are strongly related to edge density $p$ with some form of complex nonlinear relationship. To analyze the intrinsic factors influencing the PMI, it is necessary to exclude the effects of network size and edge density on path multiplicity. Thus, it inspires us to introduce the concept of the *relative path multiplicity index* (RPMI), denoted by $\widetilde{\Phi}(G)$, which is calculated by normalizing the PMI of the network $G$ with the PMI of an equivalent ER random network $G_{ER}$ with the same size and edge density as follows:

$$\widetilde{\Phi}(G) = \frac{\Phi(G)}{\Phi(G_{ER})}. \tag{3}$$

The RPMI allows us to assess how the intrinsic structural properties influence path multiplicity relative to a randomized baseline.

### Intrinsic factors of path multiplicity

To investigate the intrinsic factors influencing path multiplicity in real-world networks, we present scatter plots for the RPMI and classical network metrics along with Pearson's correlation coefficient, Spearman's correlation coefficient, and the QCR for 140 real-world networks in Fig. 2 (see the Supplementary Information for the metadata). Given that real-world networks can be disconnected, we focus only on the giant connected component of each network to ensure consistent comparisons. First, it is easy to see that these network metrics are not significantly linearly correlated with path multiplicity, where the highest Pearson's correlation coefficient is $\rho_P = 0.3164$. By inspecting Spearman's correlation coefficients, we find that the community number, global efficiency, average shortest path length, and network diameter are correlated with path multiplicity with $\rho_s > 0.6$. Furthermore, using QCR, we observe that the association between the number

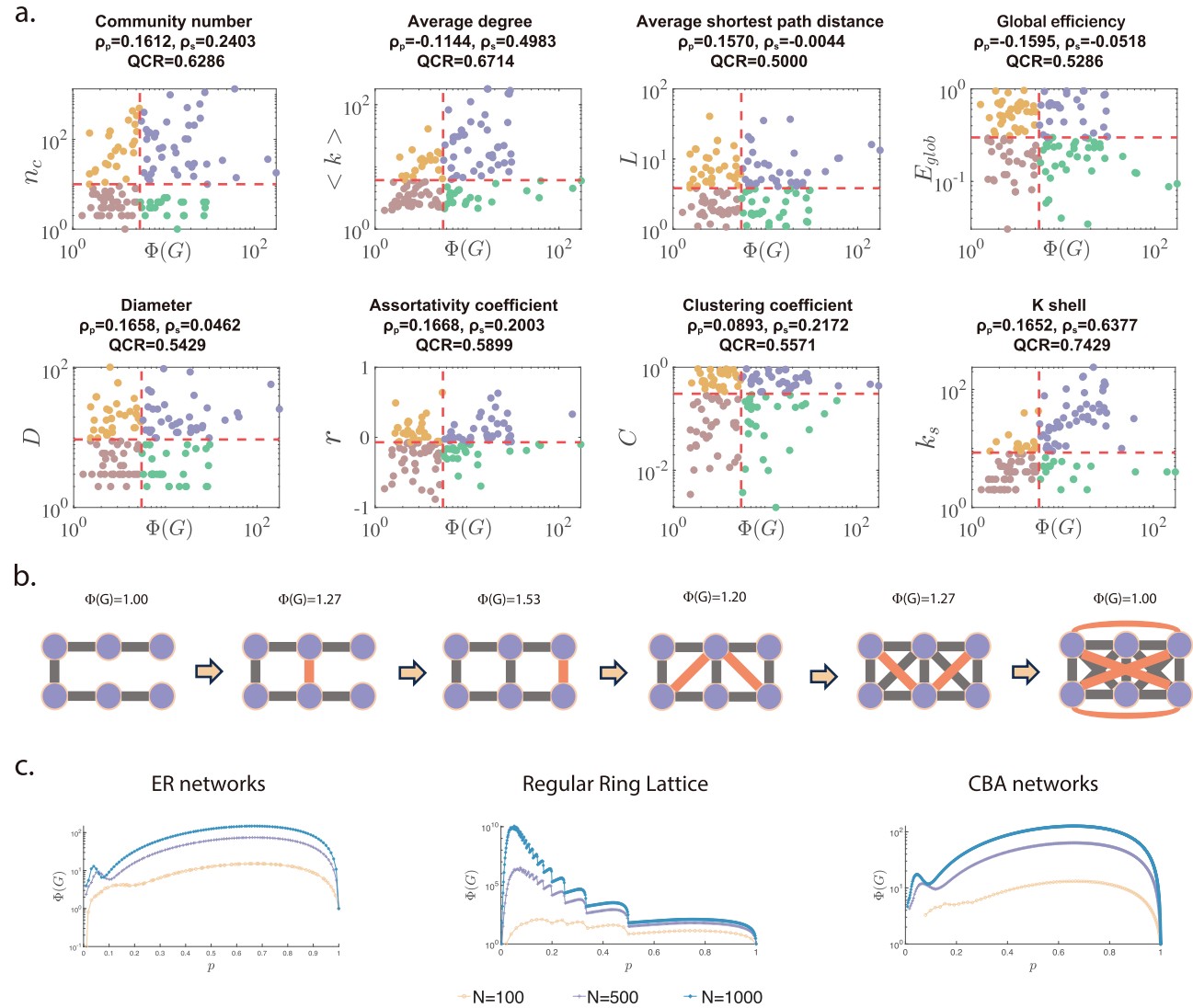

**Fig. 1 | Potential factors influencing path multiplicity. a** The scatter plots show the correlation between path multiplicity and classical network metrics in 140 real-world networks. As a reference, the median values of the PMI $\Phi(G)$ and classical network metrics are shown as dashed lines. These scatter plots reveal that the points are widely distributed throughout the plane in all four quadrants, suggesting that there are no significant correlations between path multiplicity and classical network metrics. The metadata and results for all 140 networks are provided in the Supplementary Information. The community detection algorithm used in this study is Newman's deterministic leading eigenvector modularity algorithm[43]; the specific algorithmic steps and parameter settings are detailed in the "Methods" section. **b** The example simple network is initially a chain network with an initial PMI value of 1. As the edge density increases to a fully connected network, the corresponding PMI value eventually returns to 1. **c** The PMI is plotted as a function of the edge density $p$ for three classical network models, ER random networks, regular ring lattices, and CBA scale-free networks, with different network sizes $N = 100, 500, 1000$. For all of the model networks, the PMI changes with the edge density $p$ in some kind of complex non-linear relationship forms.

of communities and path multiplicity reaches a remarkably high value of 0.9857, where the data points are mostly located in the first and third quadrants in the scatter plot shown in Fig. 2a. However, the majority of the QCR values for the other metrics are below 0.9, suggesting that the number of communities is more significantly correlated with the RPMI. In addition to the scatter plots, we display the violin plots of each network metric, which are divided into two parts on the basis of the median values of the RPMI. If a stronger correlation exists between a network metric and the RPMI, the overlap between two violin subplots should be small. As shown in Fig. 2, compared with other network metrics, the narrow overlap corresponding to the number of communities further demonstrates that the community structure[41,42] plays a key role in determining path multiplicity.

Moreover, we visualize 16 networks out of the total of 140 in Fig. 3, where the upper row presents 8 networks with high RPMI values, and the lower row shows 8 networks with lower RPMI values. In this way, we observe that networks with higher RPMI values tend to have a larger number of communities. For example, the Bio-DM-LC network with $n = 483$ ($\tilde{\Phi}(G) = 21.20$) displays highly modular structures with many well-defined communities ($n_c = 34$). In contrast, networks with lower RPMI values, such as Bio-DR-CX with $n = 3287$ ($\tilde{\Phi}(G) = 1.49$), exhibit fewer distinct communities ($n_c = 4$). This comparison clearly demonstrates that networks with higher RPMI values tend to have a greater number of communities, whereas networks with lower RPMI values are generally characterized by less modular topologies.

Considering that a specific community detection algorithm may influence the causal attribution of RPMI to community structure, we have tested classical community detection algorithms, including the Leading Eigenvector method[43], Walktrap method[44], Leiden method[45], Label Propagation method[46], Infomap method[41], and Louvain method[47]. Our experimental results are reported in the Supplementary Information section. Although these algorithms yield somewhat

### a. Community number
$\rho_p$=0.2304, $\rho_s$=0.8497, QCR=0.9857

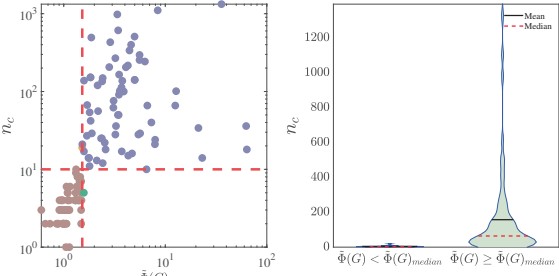

### b. Average degree
$\rho_p$=-0.1246, $\rho_s$=-0.4846, QCR=0.6714

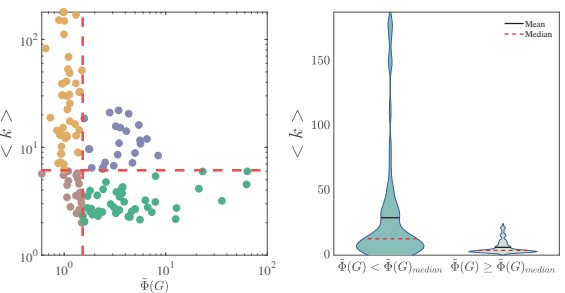

### c. Average shortest path distance
$\rho_p$=0.3007, $\rho_s$=0.7912, QCR=0.8429

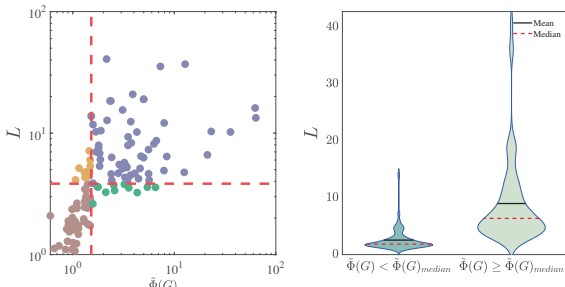

### d. Global efficiency
$\rho_p$=-0.3164, $\rho_s$=-0.8208, QCR=0.8714

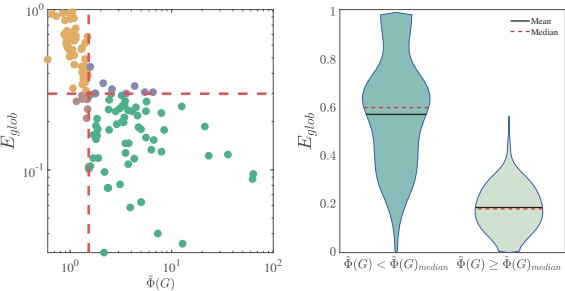

### e. Diameter
$\rho_p$=0.3106, $\rho_s$=0.8109, QCR=0.8571

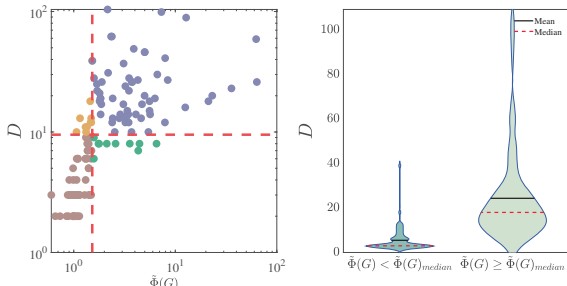

### f. Assortativity coefficient
$\rho_p$=0.1065, $\rho_s$=0.2061, QCR=0.5571

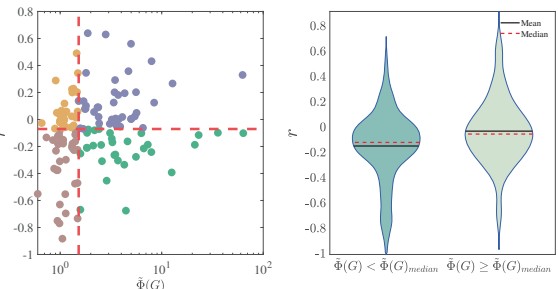

### g. Clustering coefficient
$\rho_p$=-0.0855, $\rho_s$=-0.5981, QCR=0.7429

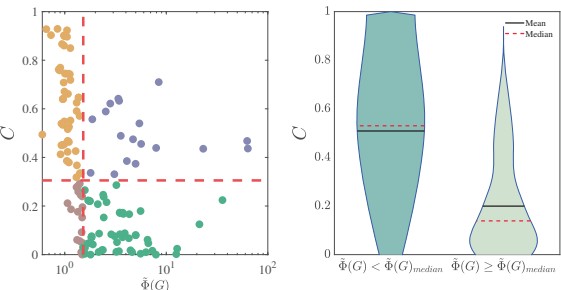

### h. K shell
$\rho_p$=-0.0765, $\rho_s$=-0.1952, QCR=0.5657

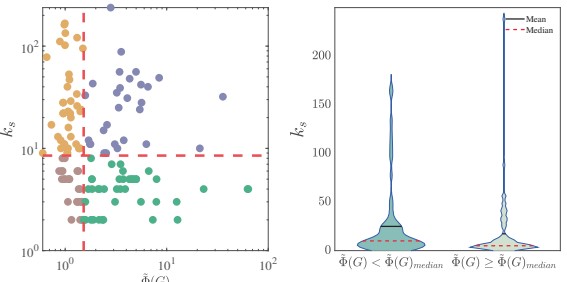

**Fig. 2 | Correlations between the relative path multiplicity and classical network metrics in 140 real-world networks.** The metadata and results for all 140 networks are provided in the Supplementary Information. The metrics include: **a** the community number $n_c$, **b** the average degree $\langle k \rangle$, **c** the average shortest path length $L$, **d** the global efficiency $E_{glob}$, **e** the diameter $D$, **f** the assortativity coefficient $r$, **g** the clustering coefficient $C$, and **h** the k-shell index $k_s$. Each **a**–**h** presents a scatter plot on the left, illustrating the correlation between the relative path multiplicity and the classical network metrics based on the given metric and their RPMI values. The networks are categorized into four quadrants, divided by the median values of the given metric and the RPMI $\tilde{\Phi}(G)$. The right-hand violin plots show the distribution, mean and median values of the corresponding network metric, which are divided into two parts based on the median value of the RPMI. If there are more overlap between two violin subplots, it suggests a weaker correlation between a network metric and the RPMI.

### a. Real-world networks with high RPMI values

Bio-DM-LC, $n_c$=34, $\tilde{\Phi}(G)$=21.20 | Rt_voteonedirection, $n_c$=66, $\tilde{\Phi}(G)$=12.54 | Road-minnesota, $n_c$=40, $\tilde{\Phi}(G)$=7.30 | Econ-mahindas, $n_c$=10, $\tilde{\Phi}(G)$=6.53

Power-685-bus, $n_c$=17, $\tilde{\Phi}(G)$=3.72 | Web-spam, $n_c$=28, $\tilde{\Phi}(G)$=3.22 | Fb-pages-food, $n_c$=62, $\tilde{\Phi}(G)$=3.08 | Email-dnc, $n_c$=18, $\tilde{\Phi}(G)$=2.58

### b. Real-world networks with low RPMI values

econ-wm3, $n_c$=5, $\tilde{\Phi}(G)$=1.57 | Bio-DR-CX, $n_c$=4, $\tilde{\Phi}(G)$=1.49 | Bn-mouse-retina-1, $n_c$=4, $\tilde{\Phi}(G)$=1.31 | Inf-USAir97, $n_c$=4, $\tilde{\Phi}(G)$=1.31

Socfb-Reed98, $n_c$=4, $\tilde{\Phi}(G)$=1.30 | Primary-school-proximity, $n_c$=4, $\tilde{\Phi}(G)$=1.10 | Bn-mouse_brain_1, $n_c$=2, $\tilde{\Phi}(G)$=0.89 | Bio-CE-PG, $n_c$=3, $\tilde{\Phi}(G)$=0.60

**Fig. 3 | Visualization of empirical networks based on RPMI values. a** Eight real-world networks with higher RPMI values. **b** Eight real-world networks with lower RPMI values. Each network visualization is colored based on its community structure, where each color represents a distinct community.

different community counts, the Spearman correlation between RPMI and the number of communities remains consistently high (all $\rho_s > 0.8$), and all the corresponding QCR values are above 0.9. These results indicate that our findings do not depend on any particular community detection algorithm; accordingly, we emphasize that our main conclusions are robust to the choice of a community detection algorithm.

### Impact of community structure on path multiplicity

To verify the relationship between path multiplicity and community structure, we implement a target-oriented edge rewiring procedure on paradigmatic model networks with 1000 nodes, namely ER random networks, Newman-Watts small-world networks (NW)[48,49], and Barabási-Albert scale-free networks (BA)[9]. As shown in Fig. 4a, we utilize the greedy principle to increase the PMI values while maintaining the same network size and edge density to observe the changes in community number during the optimization process. We find that as the PMI value increases, the community number tends to rapidly increase. For example, in the ER random network shown in the first subplot of Fig. 4a, the value of $\Phi(G)$ monotonically increases from

3.90 to 21.65, whereas the community number $n_c$ exhibits a fluctuating upward trend, increasing from 4 to 14. We also implement a target-oriented edge rewiring procedure to increase the community number and observe the changes in the PMI, as shown in Fig. 4b. Similar to the findings above, we observe that the PMI value also tends to strongly increase as the community number is optimized. These two experimental results confirm that there is a notable causal relationship between community structure and path multiplicity. As a point of comparison, we perform the above target-oriented edge rewiring experiments based on the clustering coefficient, network diameter, and assortativity. The results are provided in the Supplementary Information and show that there is no significant causal relationship between these topological properties and path multiplicity.

### Community-based network model

Inspired by the above results, we next develop a Tribal Scale-Free (TSF) model that can reproduce the hesitant-world features of real-world networks. The TSF model generates networks with hierarchical and modular structures by creating scale-free subnetworks and

## a. Variation of $n_c$ values with incremental PMI optimization

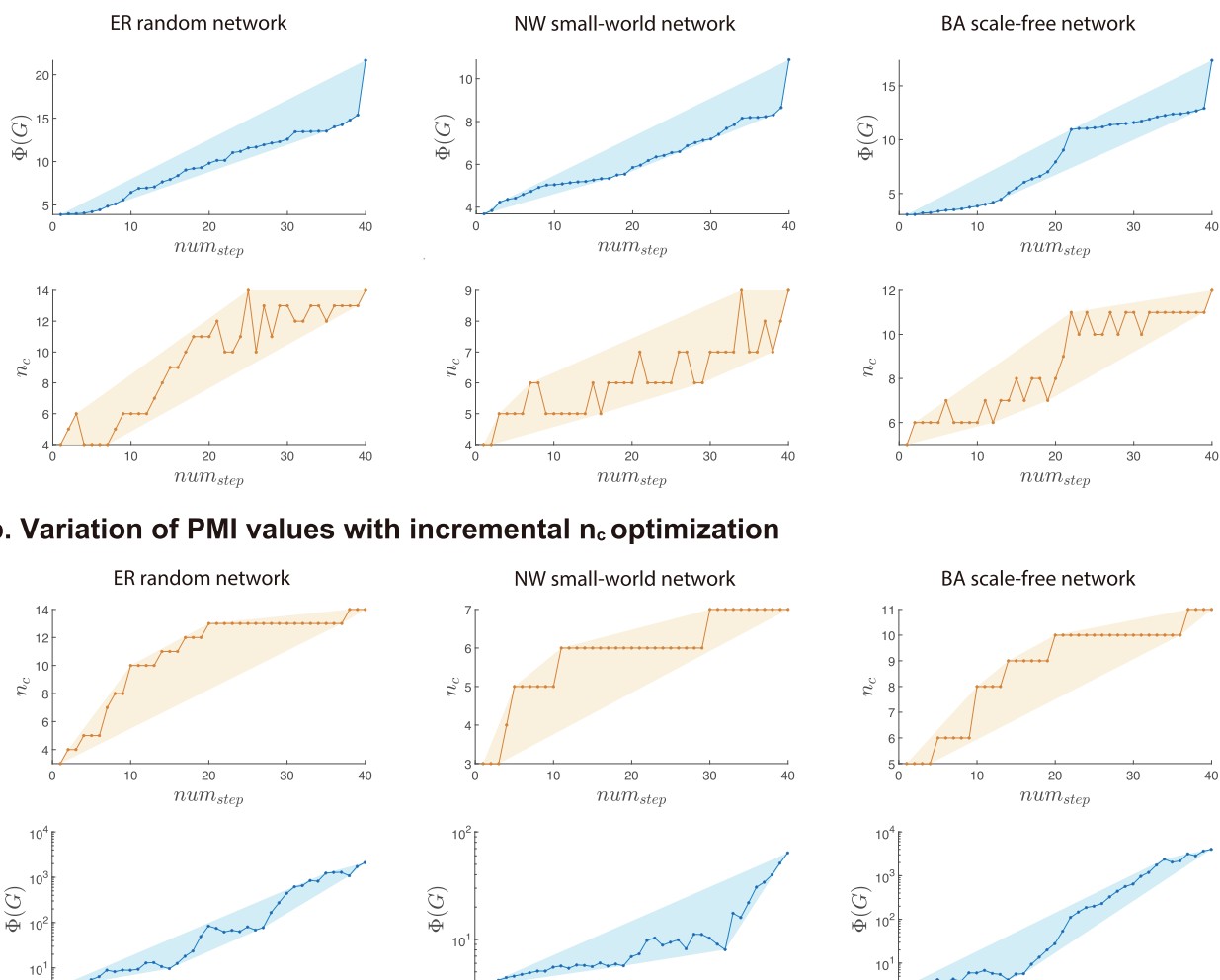

## b. Variation of PMI values with incremental $n_c$ optimization

**Fig. 4 | Target-oriented edge rewiring in three types of model networks. a** The panels depict the variation in the $n_c$ values with incremental optimization of the PMI through target-oriented edge rewiring. As the PMI increases, the number of detected communities $n_c$ tends to increase overall. **b** The panels show the variation in the PMI values with incremental optimization of the community number $n_c$ through target-oriented edge rewiring. In this scenario, edge rewiring is directed toward increasing $n_c$, resulting in a trend exhibiting rising PMI values. Shaded regions indicate simple envelopes for visual guidance, not uncertainty or confidence intervals.

interlinking them through a controlled number of intercommunity edges. The specific generation rules are described in the "Methods" section. To verify the effectiveness of the proposed TSF model, we compare the real PMA distributions $P(h)$, the maximum PMA values and the PMI values $\Phi(G)$ with the synthetic results based on four typical model networks, i.e., the TSF network, the ER random network, the NW small-world network, and the BA scale-free network. As shown in Fig. 5a, we find that the TSF model reproduces the hesitant-world features observed from real-world networks much better than the other model networks. For example, in the Bio-SC-LC network with $\Phi(G) = 21.50$, where the maximum PMA value is 7189, the corresponding TSF network has a PMI value of 21.46, with a maximum PMA value of 7201. However, the PMI values are 4.99, 6.79 and 8.42 in the ER random network, the NW small-world network and the BA scale-free network, respectively, and the maximum values of the PMA are 185, 802 and 398, respectively. In addition, in Fig. 5b, we present the PMI values of representative real-world networks along with the PMI values of the reference model networks with the same network size and similar edge density. We find that the TSF model can better reproduce the PMI values observed from real-world networks and thus can

capture the hesitant-world feature, whereas the other three classical network models significantly deviate from the empirical PMI values.

## Discussion

Following the introduction of the concept of path multiplicity, recent studies have revealed that empirical networks display not only the well-known "small-world" effect, but also an intriguing "hesitant-world" feature, accompanied by a universal power-law governing path multiplicity. In this study, we investigated the mechanisms underlying path multiplicity. We introduced the concept of the relative path multiplicity index (RPMI) to explore the intrinsic factors that influence path multiplicity by excluding the effects of network size and edge density. We uncovered a robust positive correlation between the number of communities and the RPMI by employing comprehensive correlation analysis methods: Pearson correlation, Spearman correlation, and QCR. This reveals that networks with more communities exhibit significantly greater path multiplicity. Then, we verified this observation by using the target-oriented edge rewiring procedure, where a higher path multiplicity was associated with a greater number of communities, and vice versa. The underlying mechanism can be

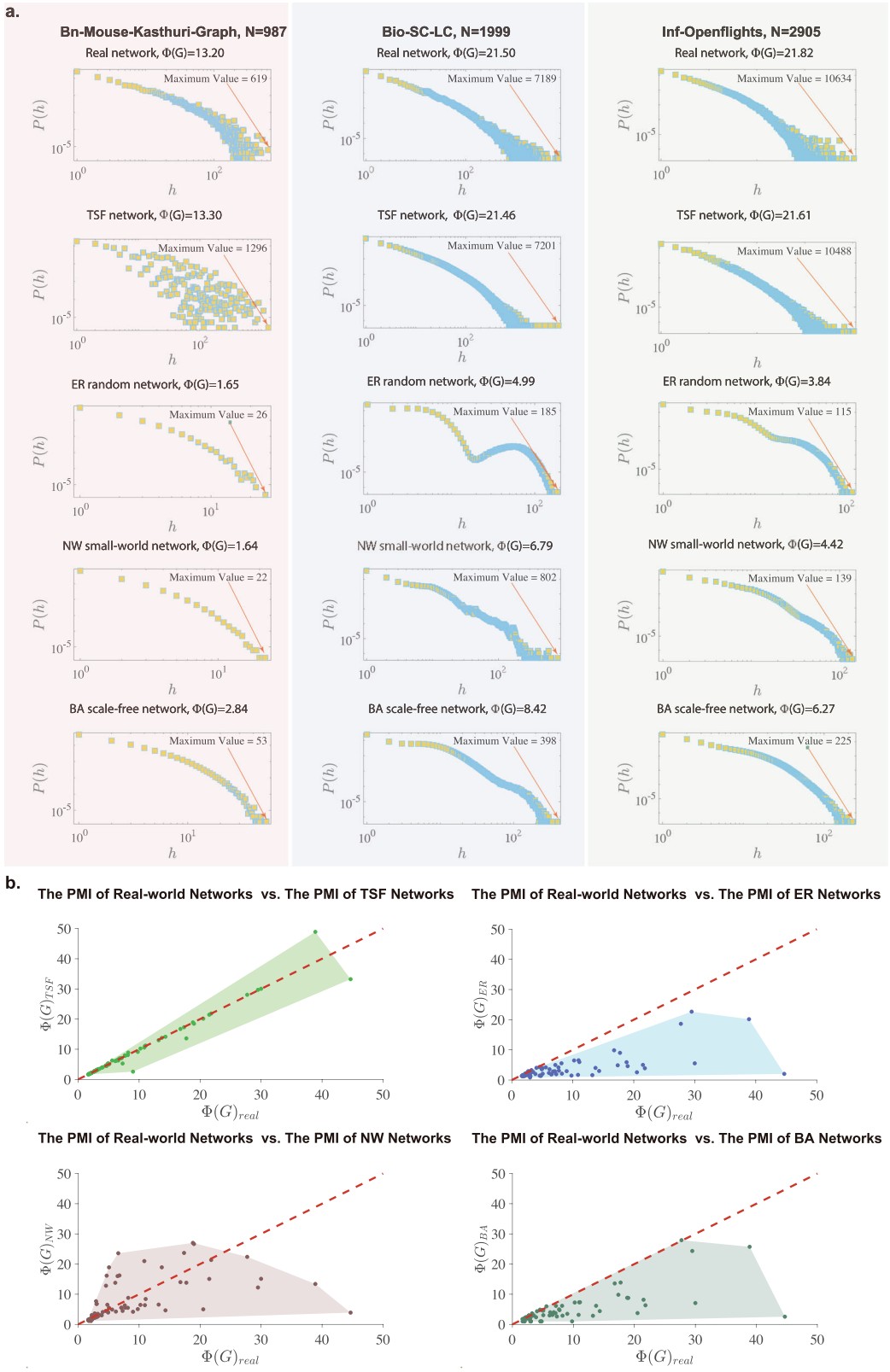

**Fig. 5 | Verification of the effectiveness of the proposed TSF model.**
**a** Comparisons of real PMA distributions, maximum PMA values and PMI values with the synthetic results are conducted with four typical model networks, i.e., the TSF network, ER random network, NW small-world network, and BA scale-free network. Compared with real-world networks, classical model networks have the same number of nodes and similar edge densities. **b** Scatter plots for the $\Phi(G)$ of representative real-world networks and the corresponding model networks $G_{TSF}$, $G_{ER}$, $G_{NW}$, and $G_{BA}$ with the same network size and similar edge densities. The dashed line, with the darker shaded area, denotes a reference line with a slope of 1.

interpreted as an interface-driven effect: intercommunity edges constitute an effective cut set, and therefore, path multiplicity between modules necessarily passes through boundary nodes and bridges; when multiple boundary-equivalent intramodular segments and cross-community links are length-equivalent, their combinations multiply, leading to a sharp increase in the shortest path counts. Finally, inspired by the above results, we designed the TSF model to replicate the hesitant-world feature observed from empirical networks. By comparing the TSF model with typical model networks such as the ER random network, the NW small-world network and the BA scale-free network, we demonstrated that the TSF model can much better reveal the power-law distribution of the PMA, the maximum values of the PMA, and the PMI values observed from real-world networks. The hesitant-world feature, characterized by an abundance of shortest paths, presents challenges and opportunities in the study of network science. It could facilitate valuable applications in diverse fields—including urban infrastructures, communication networks, neuroscience, and artificial intelligence—thereby driving innovation. Our work represents a foundational step that will enable deeper exploration of path multiplicity in the future.

This study identifies community structure as the key driver of path multiplicity in unweighted networks. A crucial next step is to extend this analysis to weighted networks—the structure of most real-world systems, from transportation to social interactions. Edge weights fundamentally alter the shortest path landscape: the condition for multiplicity (multiple paths with an identical total weight) becomes mathematically strict, often reducing many shortest paths to a single optimal route. This shift reveals an important gap and opportunity, as practical applications often concern near-optimal alternatives rather than perfectly equivalent paths. To bridge this gap, future work could generalize the concept by introducing such a tolerance parameter $\epsilon \geq 0$, defining the paths within $(1 + \epsilon)$ of the minimum weight as "tolerant shortest paths." Analyzing the multiplicity of such paths would clarify how network architecture—particularly the community structure—supplies robust routing options under realistic conditions, directly linking our topological findings to challenges in resilient infrastructure, adaptive traffic routing, and the analysis of neural or social networks with continuous link strengths.

Moreover, this paper only draws evidence from empirical networks and simulations; a fuller theoretical account remains to be developed. Promising high-impact research directions could include an analytical relationship between path multiplicity and structural parameters, spectral approaches to bound the path multiplicity, and cut/flow arguments via Menger's theorem that link cross-community cut size and boundary redundancy.

## Methods

### Fast algorithm for counting shortest paths

The shortest path lengths between node pairs can be calculated by using the classical Floyd algorithm[50], which has a time complexity of $O(N^3)$. However, this algorithm does not provide the count of shortest paths between node pairs. Although Breadth-First Search or Depth-First Search can identify all paths between a pair of nodes in $O(N^2)$ time, calculating the number of shortest paths for all pairs of nodes with these methods increases the complexity to $O(N^4)$.

Path multiplicity focuses on the number of shortest paths between node pairs rather than on the nodes traversed. In this study, we implement a fast algorithm to calculate both the number and length of shortest paths between all node pairs simultaneously. The procedure is described as follows.

**Step 0**. Initialize the Path Multiplicity Matrix $H = \{h_{ij}\}_{N \times N}$ and the Path Length Matrix $L = \{l_{ij}\}_{N \times N}$, setting $h_{ij} = h_{ji} = 0$ and $l_{ij} = l_{ji} = 0$. Define $k = 1$.

**Step 1**. If $h_{ij} = h_{ji} \neq 0$ for all $i \neq j$, terminate the algorithm. Otherwise: (a) Compute the transition matrix $T = \{t_{ij}\}_{N \times N}$, where $t_{ij} = 0$ if $h_{ij} \neq 0$,

and $t_{ij} = 1$ if $h_{ij} = 0$. (b) Update $H$ to $H = H + T \odot A^k$, where $\odot$ denotes the Hadamard product. (c) Update $L$ to $L = L + k \cdot (T \odot J)$, where $J$ is an $N \times N$ all-ones matrix.

**Step 2**. Increment $k$ by 1 and return to Step 1.

### Quadrant count ratio

To quantify the monotonic relationship between relative path multiplicity and a given network metric, we employ the improved QCR, a non-parametric measure of association derived from quadrant analysis. The QCR provides a coarse-grained measure of association by dichotomizing each variable at a central location (typically the median) and quantifying how often paired observations fall in the same corresponding half. Specifically, QCR is computed by splitting each variable into "low" and "high" groups using its median (or another robust central threshold) and then calculates the proportion of paired samples that are concordant—i.e., simultaneously high-high or low-low—among all pairs.

Given paired observations $\{(x_i, y_i)\}_{i=1}^{N}$ for two variables $X$ and $Y$, define $\tilde{x}$ and $\tilde{y}$ as the sample medians of $\{x_i\}$ and $\{y_i\}$, respectively. The plane is divided by the vertical line $x = \tilde{x}$ and the horizontal line $y = \tilde{y}$ into four quadrants:

- Quadrant I ($x \geq \tilde{x}, y \geq \tilde{y}$);
- Quadrant II ($x < \tilde{x}, y > \tilde{y}$);
- Quadrant III ($x \leq \tilde{x}, y \leq \tilde{y}$);
- Quadrant IV ($x > \tilde{x}, y < \tilde{y}$).

We count the number of points $n_{\text{I}}, n_{\text{II}}, n_{\text{III}}, n_{\text{IV}}$ that fall in each quadrant. If a positive monotonic relationship is hypothesized, points in Quadrant I and Quadrant III share the same sign of deviation from the medians and are deemed consistent, whereas points in Quadrants II and Quadrant IV are inconsistent. Thus, the QCR is defined as

$$\text{QCR}_+ = \frac{n_{\text{I}} + n_{\text{III}}}{N}, \tag{4}$$

which takes values in [0, 1]. A value close to 1 indicates that most observations lie in the first or third quadrant, implying a strong positive association, while a value near 0.5 suggests no directional preference. When a negative monotonic association is hypothesized, we instead treat Quadrants II and IV as consistent and define

$$\text{QCR}_- = \frac{n_{\text{II}} + n_{\text{IV}}}{N}. \tag{5}$$

Consequently, a higher QCR indicates stronger association, because a larger fraction of observations exhibit consistent co-movement in the same direction relative to the central location (whereas QCR near 0.5 suggests weak or no systematic alignment, and values below 0.5 indicate predominantly discordant pairing).

### Community detection via leading eigenvector modularity

Non-overlapping communities are detected by maximizing the modularity of the undirected graph $G(V, E)$ in a deterministic spectral framework. Let $A(G) = (a_{ij})_{N \times N}$ denote the adjacency matrix defined above, $k_i = \sum_j a_{ij}$ the degree of node $v_i$, and $M = |E|$ the number of edges, so that $2M = \sum_{i,j} a_{ij}$. Define the modularity matrix

$$B = A(G) - \frac{\mathbf{k}\mathbf{k}^{\top}}{2M}, \tag{6}$$

where $\mathbf{k} = (k_1, \ldots, k_N)^{\top}$.

For a bipartition encoded by $\mathbf{s} \in \{\pm 1\}^N$, the modularity can be written as

$$Q = \frac{1}{4M} \mathbf{s}^{\top} B \mathbf{s}. \tag{7}$$

Let $\mathbf{u}_1$ be the leading eigenvector of $B$. An initial split is obtained by the sign rule $s_i = \text{sign}(u_{1i})$. A single-vertex refinement is then applied: iteratively move the node that yields the largest positive modularity gain $\Delta Q$, and repeat until no further improvement is possible ($\Delta Q \leq 0$).

To obtain more than two groups, recursion is applied to any accepted vertex subset $U \subseteq V$ (a current community). The net modularity change from subdividing $U$ is evaluated with the generalized modularity matrix

$$B_{ij}^{(U)} = B_{ij} - \delta_{ij} \sum_{\ell \in U} B_{i\ell}, \ i, j \in U, \tag{8}$$

where $\delta_{ij}$ is the Kronecker delta, and the same spectral-plus-refinement steps are applied within $U$. A subdivision is accepted only if it increases $Q$ ($\Delta Q > 0$); recursion stops when all current groups are indivisible under this criterion.

Matrix-vector products with $B$ are evaluated as

$$B\mathbf{x} = A(G)\mathbf{x} - \frac{\mathbf{k}^\top \mathbf{x}}{2M}\mathbf{k}, \tag{9}$$

so each multiply costs $O(M + N)$. Using the power or Lanczos method for $\mathbf{u}_1$, the overall cost is typically $O(N^2)$ on sparse graphs. The procedure is deterministic given $A(G)$ (up to a global sign flip of $\mathbf{u}_1$, which does not affect the partition). The algorithm returns the final community labels $\{c_i\}_{i=1}^N$ and the achieved modularity $Q$. In practice, we used the Brain Connectivity Toolbox function *modularity_und*[51] with the resolution parameter fixed at its default value $\gamma = 1$ for all community detection analyses.

## Target-oriented edge rewiring procedure

A greedy algorithm for target-oriented edge rewiring is designed to iteratively optimize a specific network metric while preserving structural properties such as the network size and edge density. The algorithm is operated by modifying the edge structure of a network to enhance a target metric and simultaneously observe its impact on the associated metric, providing insight into the integration of these properties. The procedure involves the following steps:

**Step 1: Initialize the network metrics.** Begin with a given connected network $G^0$ of size $N$ and edge density $p$, which serves purely as a rewiring seed for subsequent optimization. Some initial networks may already have high baseline metric values (e.g., community number or clustering coefficient), potentially biasing the early search toward local optima; accordingly, the first recorded iteration reflects the best outcome of the initial rewiring trials rather than the original network, and its metric value can be higher or lower than the initial one without affecting subsequent optimization.

**Step 2: Identify potential edge rewiring.** At iteration $t$, starting from the current network $G^{(t)}$ with $|E| = m$ edges, generate a set of candidate rewiring trials $\mathcal{R}^{(t)} = \{r_1^{(t)}, \ldots, r_Q^{(t)}\}$. Each trial $r_q^{(t)}$ consists of a sequence of admissible edge rewirings applied to $G^{(t)}$, with a random rewiring budget $b$ per trial, where $1 \leq b \leq m$. Each trial can modify multiple edges while strictly preserving $N$, $m$, and network connectivity; any tentative swap that would violate these structural constraints is rejected and reverted.

**Step 3: Select the optimal edge rewiring with tie handling.** For each trial $r_q^{(t)} \in \mathcal{R}^{(t)}$, compute the target metric on the resulting network and identify the trial(s) that achieve the maximum target metric value, i.e.,

$$\mathcal{Q}_{\max}^{(t)} = \arg\max_{1 \leq q \leq Q} f\left(G^{(t)} \circ r_q^{(t)}\right), \tag{10}$$

where $G^{(t)} \circ r_q^{(t)}$ denotes applying trial $r_q^{(t)}$ to $G^{(t)}$. If multiple trials attain the same maximum value (including the case where the maximum equals the target value of the previous accepted network), randomly

select one trial $q^\star \in \mathcal{Q}_{\max}^{(t)}$ and use the corresponding rewired network as the next state.

**Step 4: Update the network and metrics.** Apply the selected trial to obtain the updated network $G^{(t+1)}$. Recalculate both the target and associated metrics on $G^{(t+1)}$. Record these values to track the optimization progress, noting that the first recorded iteration reflects the best candidate network from the initial trials rather than the original network.

**Step 5: Termination check.** Repeat Steps 2-4 until a predefined number of valid rewired networks has been recorded, where "valid" means the rewired network satisfies the structural constraints (e.g., fixed $N$ and $m$, and any additional constraints such as connectivity).

**Step 6: Analyze the results.** Once the optimization is complete, examine the relationship between the optimized target metric and the associated metric. For example, after maximizing $\Phi(G)$, observe how the community number $n_c$ changes across iterations.

This algorithm enables a systematic exploration of how network metrics influence each other through structural modifications. By maintaining the network size and edge density, the approach ensures that the observed trends are solely attributable to the optimization process, rather than external structural changes. The experimental results, such as those shown in Fig. 4, demonstrate the effectiveness of this method in terms of revealing the causal relationship between community structure and path multiplicity.

## Tribal Scale-Free (TSF) network model

The TSF model is a generative model designed to capture the hierarchical and modular features typical of real-world networks, adding the separation of distinct communities (cf. the "connected caveman model"[7]). The TSF model builds a network by integrating intra-community scale-free subnetworks with controlled intercommunity connections, allowing for an examination of the relationship between modularity and path multiplicity. The detailed mathematical formulation of the TSF model is presented below, with a step-by-step description of its generation process.

**Step 1: Initialization.** Start with a total network size $N$ and a specified number of communities $n_c$. The $N$ nodes are randomly assigned to $n_c$ distinct communities, representing different "tribes". Let $C_i$ denote the $i$-th community, where $i \in 1, 2, \ldots, n_c$. The size of each community $|C_i|$ is determined such that

$$\sum_{i=1}^{n_c} |C_i| = N. \tag{11}$$

**Step 2: Generation of scale-free subnets.** Within each community $C_i$, a scale-free network is generated by using the the Goh-Kahng-Kim (GKK) static scale-free network model[52]. The GKK model constructs a random graph by assigning each vertex $v \in C_i$ a weight $p_v \propto v^{-\alpha_i}$ with $\alpha_i \in [0, 1)$, then repeatedly selecting two different vertices according to the normalized weights and adding an edge between them unless one exists already. By tuning $\alpha_i$, the degree distribution follows a power-law characterized by an exponent $\lambda_i$ via $\lambda_i = (1 + \alpha_i)/\alpha_i$. The resulting degree distribution adheres to the following form:

$$P(k) \sim k^{-\lambda_i}, \ \text{for } k \geq k_{\min}. \tag{12}$$

Other approaches for generating scale-free networks can also be adopted, such as the BA scale-free model and configuration model (CM)[53].

**Step 3: Intercommunity edge allocation.** Once scale-free intra-community subnets are generated, proceed with intercommunity

edge allocation. Given a predefined number of intercommunity edges $n_e$, two distinct communities, $C_i$ and $C_j$, are selected, and then $n_e$ edges are created between them. For subsequent connections, an unconnected community $C_m$ is chosen uniformly at random, and this community is linked with $n_e$ edges to any of the already connected communities. This process is repeated iteratively until all of the communities are connected, ensuring the formation of a single connected component throughout the network.

**Step 4: Construction and finalization of the network.** The TSF network $G = (V, E)$ consists of $|V| = N$ nodes and a set of edges $E$, combining the intracommunity edges $E_{intra}$ and the intercommunity edges $E_{inter}$ such that $E = E_{intra} \cup E_{inter}$. The total number of edges in the network is expressed as follows:

$$|E| = \sum_{i=1}^{n_c} |E_{intra}^i| + n_e \cdot (n_c - 1). \tag{13}$$

Notably, to keep the model simple, the proposed TSF model assumes a common scaling exponent across communities. In practice, scaling exponents can differ between communities in real-world networks[54–57], and future extensions of the TSF model can therefore allow heterogeneous, community-specific scaling exponents rather than enforcing a single global value.

Furthermore, the TSF model can effectively capture the modular and hierarchical nature of complex networks by combining scale-free substructures within communities and interconnecting them through a controlled number of intercommunity edges. By adjusting the parameters $n_c$ (number of communities) and $n_e$ (number of intercommunity edges), the model facilitates a systematic exploration of the impact of community structure on network properties such as the path multiplicity and overall network efficiency.

## Data availability
The network datasets used in this study are publicly available at the Network Repository (https://www.networkrepository.com) and can be freely downloaded without restriction. All data supporting the findings of this study are available in the Supplementary Information. Source data are provided with this paper.

## Code availability
The custom codes used in this study are available on GitHub(https://github.com/gituserbnu/path-multiplicity.git) and archived on Zenodo[58].

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

## Acknowledgements

J.W. acknowledges support from the National Natural Science Foundation of China (72571031), and the Innovation Teams Project in Ordinary Universities of Guangdong Province (2024KCXTD050). Y.D. acknowledges support from the National Natural Science Foundation of China (72201035), and the Natural Science Foundation of Guangdong Province (2025A1515011586). X.L. acknowledges support from the National Natural Science Foundation of China (72025405, 72421002, 92467302), the Brain Science and Brain-like Intelligence Technology—National Science and Technology Major Project (2025ZD0215700), and the Major Program of Xiangjiang Laboratory (24XJJCYJ01001). D.L. acknowledges support from the National Natural Science Foundation of China (72225012, 72288101). We sincerely appreciate the valuable computational resources provided by the School of Systems Science at Beijing Normal University (https://sssdata.bnu.edu.cn) for our research.

## Author contributions

J.W. conceived this study; Y.D., J.W., X.L. and D.L. designed the research framework; Y.D., J.W., X.L. and D.L. performed the research; Y.D. and J.W. analyzed the data; Y.D., J.W., X.L., P.H., D.L., Z.D., G.C. and J.K. wrote the paper. All authors contributed and provided critical feedback and helped shape the research, analysis and manuscript.

## Competing interests

The authors declare no competing interests.
