## [Transparent Peer Review file · Nature Communications]

Community structure unveils the path multiplicity in complex networks

Corresponding Author: Professor Jun Wu

Version 0:

Reviewer comments:

Reviewer #1

(Remarks to the Author)

This paper offers an insightful contribution to network science by investigating the underlying mechanisms behind path multiplicity in complex networks, specifically, the observed heterogeneity in the number of shortest paths between node pairs. The authors compellingly identify community structure as a key driver of this phenomenon, a connection that has thus far remained largely unexplored. A major strength of the paper lies in its introduction of the relative path multiplicity metric, which effectively normalises for network size and edge density, allowing for a more solid analysis. The study is further strengthened by the authors' systematic comparison of classical network metrics and their correlations with path multiplicity, where community structure stands out as the dominant factor. The use of targeted edge rewiring experiments to validate the relationship between community structure and path multiplicity is convincing. Overall, this work is a significant and original contribution, advancing our understanding of structural features in complex networks and opening up promising avenues for future research.

As minor, optional comments, the authors could emphasise connections between the notion of cut and community detection, as the cut size is related to the number of independent paths existing between nodes.

Reviewer #2

(Remarks to the Author)

General assessment:

The manuscript addresses an understudied yet important network property - path multiplicity - and demonstrates its empirical relevance. The analysis is strengthened using a large and diverse dataset of real-world networks, which supports the broad applicability of the findings. Moreover, the edge-rewiring experiments provide compelling evidence linking community structure to path multiplicity. The proposed TSF generative model successfully captures the observed empirical behavior, outperforming standard random-graph models and opening avenues for further theoretical work. Overall, these contributions advance our understanding of structural redundancy and resilience in complex networks.

On the other hand, it is increasingly recognized that purely structural metrics often have limited explanatory power without a clear link to network functions or dynamics (Gates & Rocha, 2016 for network control; Hines et al., 2010; Pahwa et al., 2012 for power grids; Zhang et al., 2015 for transportation systems). Therefore, introducing a new structural metric — especially one limited to unweighted networks — requires a strong justification of its utility beyond existing descriptors. To strengthen the paper's impact, the authors are encouraged to further elaborate on the practical importance of path multiplicity. This could be achieved by discussing or demonstrating its role in a functional or dynamic context, such as network routing, spreading processes, or resilience to failures, showing how the metric offers truly novel insights not captured by classical measures.

Detailed comments:

1. A discussion/analysis on how the conclusions might change for weighted and/or directed edges would be important.
2. I would be hesitant using the term "path hesitation" to convey the concept of path multiplicity.

3. The causal attribution of RPHI to community structure is dependent on the specific community detection algorithm used. The authors should specify which algorithm and resolution parameter were employed.
4. The quadrant-consistency rate (QCR) is central to the analysis but does not appear to be a standard metric in network science. The authors should provide a justification for using QCR.
5. Figure 1 is unfortunately confusing for an introductory figure. For example, while its caption promises to show a “Relationship between the path multiplicity and classical network metrics,” the data and the demonstration do not reveal any relationship.
6. Typos: “Path multiplicity <has> profound implications for network efficiency and route decision-making” – on page 2 and some others.
7. Could a high RPHI also arise from a core-periphery structure? For example, in a network with few communities but a very densely connected core.
8. The TSF model assumes a single power-law exponent, λ , across all communities. Given that real-world networks often exhibit community-specific exponents (Nature 435, 814-818, 2005, Phys. Rev. E 94, 012302, 2016), a discussion on the implications of this simplifying assumption and potential model extensions would be welcome.
9. The rewiring method in Sec. 4.2 preserves the degree sequence and global connectivity but does not constrain other topological properties like clustering, diameter, or assortativity. Since these metrics could also influence PHI, it would be helpful to report their evolution during the optimization process.
10. Perhaps the paper’s theoretical contribution could also be strengthened. Have the authors considered deriving an analytical relationship between path multiplicity and community structure parameters, for instance within a stochastic block model? Another valuable approach would be to use spectral methods to obtain rigorous theoretical bounds on path multiplicity.

Reviewer #3

(Remarks to the Author)

Version 1:

Reviewer comments:

Reviewer #1

(Remarks to the Author)

Thank you for updating your manuscript following my suggestions.

Reviewer #2

(Remarks to the Author)

I thank the authors for considering my previous comments and providing detailed feedback. While the manuscript has certainly improved, I am afraid that the authors have not fully addressed the main concerns. Again, the concept of path multiplicity is definitely interesting. However, I do not think the overall quality and rigor of the manuscript, even after the revision, meet the standards for Nature Communications. I have the following lingering concerns:

1. Previously, I noted that introducing a new structural metric, particularly one restricted to unweighted networks, requires strong justification. In response, the authors provided a discussion on the relevance of path multiplicity, which feels brief and generic. No references are cited to support this discussion. No justification of the necessity (or un必要性) of considering edge weights is given. The meanings of “structural proxy” and “increasing ambiguity” remain vague in this context. The transition to the statement that “...the mechanisms behind path multiplicity remain largely unexplored...” also feels abrupt and disrupts the flow.
2. I also noted that the correlation between RPHI and community structure is likely dependent on the specific community detection algorithm used, which the authors agreed: “...We agree that the causal attribution of RPHI to community structure is dependent on the specific community detection algorithm used...” However, this specification or limitation is not mentioned in the manuscript, which risks exaggerating the paper’s conclusion. While the authors claim that “...main conclusions are robust to the choice of community detection algorithm...” and stated in their reply that they tested algorithms such as Louvain, Leiden, and Infomap, no results are given in the Main Text, Methods, or Supplementary Information. Given the authors’ statement that “...this paper primarily focuses on the impact of community structure on path multiplicity,” it seems biased to report results for only a single algorithm. I urge the authors to include these comparative results to substantiate

their claim.

3. I asked the authors to justify why QCR was chosen. They responded by "...Its value lies in complementing standard coefficients such as Pearson's correlation coefficient and Spearman's correlation coefficient: when associations are monotonic but nonlinear, masked by heavy tails or outliers, or split across clusters that depress the correlation coefficient, QCR can still reveal systematic alignment between variables..." First of all, Spearman's correlation coefficient already takes nonlinear monotonicity into account, so this argument is not convincing. I am asking for a fundamental reason why QCR is the appropriate metric for this system: why should QCR be high, and what physical or structural insight does that convey? Without a deeper justification, the use of QCR appears superficial and selective.

4. I asked the authors to reconsider the terminology "path hesitation." The authors insisted on keeping it, but the term remains confusing and has not been properly motivated or defined. For example, the phrase "hesitant-world" abruptly appears on page 3 without any definition regarding what it implies physically or structurally. If the authors wish to retain this terminology, it must be clearly spelled out.

Overall, I still believe the paper will be of interest to specific readers (myself included). However, the current quality of the presentation is concerning. I hope the authors take these comments seriously and undertake a more substantial revision.

Reviewer #3

(Remarks to the Author)

Version 2:

Reviewer comments:

Reviewer #2

(Remarks to the Author)

The paper has been substantially revised. I have no further comments and would like to thank the authors for considering the suggestions.

Reviewer #3

(Remarks to the Author)

Point-by-point Responses to Reviewers' Comments

✧ Reviewer #1:

General assessment: This paper offers an insightful contribution to network science by investigating the underlying mechanisms behind path multiplicity in complex networks, specifically, the observed heterogeneity in the number of shortest paths between node pairs. The authors compellingly identify community structure as a key driver of this phenomenon, a connection that has thus far remained largely unexplored. A major strength of the paper lies in its introduction of the relative path multiplicity metric, which effectively normalises for network size and edge density, allowing for a more solid analysis. The study is further strengthened by the authors' systematic comparison of classical network metrics and their correlations with path multiplicity, where community structure stands out as the dominant factor. The use of targeted edge rewiring experiments to validate the relationship between community structure and path multiplicity is convincing. Overall, this work is a significant and original contribution, advancing our understanding of structural features in complex networks and opening up promising avenues for future research.

Response: We sincerely thank the reviewer for this positive and encouraging overall assessment of our work, and we have carefully revised the manuscript to further clarify the motivation, methodology, and implications of our findings.

Comment 1: As minor, optional comments, the authors could emphasise connections between the notion of cut and community detection, as the cut size is related to the number of independent paths existing between nodes.

Response 1: We sincerely thank the reviewer for the insightful comment. As you mentioned, there is indeed a connection between the notion of cut and community detection. Our study identifies community structure as the key factor associated with path multiplicity. The underlying mechanism might be interpreted as an interface-driven effect: intercommunity edges constitute an effective **cut set**, and therefore, path multiplicity between modules necessarily passes through boundary nodes and bridges; when multiple boundary-equivalent intramodular segments and cross-community links are length-equivalent, their combinations multiply, yielding a sharp increase in the shortest path counts.

In the revised version, we have added explanations in the *Abstract* and the first paragraph of the *Discussion* as follows.

Abstract

Networks with complex topologies describe numerous natural and social systems. The paths that connect nodes and edges are fundamental components in network science and are important for exploring network connectivity, communication efficiency, and load balancing. Recent studies on path multiplicity have shown extreme heterogeneity in the number of shortest paths between pairs of nodes in real-world complex networks. However, the mechanism underlying this phenomenon remains unexplored. Here, we reveal that the community structure is responsible for path multiplicity. We first introduce the concept of relative path multiplicity to explore the intrinsic factors that influence path multiplicity by excluding the effects of network size and edge density; on this basis, we then further explore the relationship between classical network metrics and relative path multiplicity. We find that compared with other metrics, the community structure is more strongly correlated with path multiplicity. Furthermore, through target-oriented edge rewiring experiments, we verify the indispensable connection between path multiplicity and the community structure. **The underlying mechanism can be interpreted as an interface-driven effect that yields a sharp increase in the number of shortest paths.** Inspired by these observations, we propose a novel network model based on a tribal structure that can be used to successfully reproduce phenomena observed in the real world. Our findings improve the understanding of network structures, with significant potential applications in network design and optimization.

Discussion

Recent studies have revealed that empirical networks display not only the well-known small-world property but also an intriguing hesitant-world feature, accompanied by a universal power-law governing path multiplicity. In this study, we investigated the mechanisms underlying path multiplicity. We first introduced the concept of the relative path hesitation index (RPHI) to explore the intrinsic factors that influence path multiplicity by excluding the effects of network size and edge density. We have uncovered a robust positive correlation between the number of communities and the RPHI through four-quadrant consistency analysis, revealing that networks with more communities exhibit significantly greater path multiplicity. Then, we verified this observation by using the target-oriented edge rewiring procedure, where a higher path multiplicity was associated with a greater number of communities, and vice versa. **The underlying mechanism can be interpreted as an interface-driven effect: intercommunity edges constitute an effective cut set, and therefore, path multiplicity between modules**

necessarily passes through boundary nodes and bridges; when multiple boundary-equivalent intramodular segments and cross-community links are length-equivalent, their combinations multiply, leading to a sharp increase in the shortest path counts. Finally, inspired by the above results, we designed the Tribal Scale-Free (TSF) model to replicate the hesitant-world feature observed from empirical networks. By comparing the TSF model with typical model networks such as the ER random network, the NW small-world network and the BA scale-free network, we demonstrated that the TSF model can much better reveal the power law distribution of the PHA, the maximum values of the PHA, and the PHI values observed from real-world networks.

✧ **Reviewer #2:**

General assessment: The manuscript addresses an understudied yet important network property - path multiplicity - and demonstrates its empirical relevance. The analysis is strengthened using a large and diverse dataset of real-world networks, which supports the broad applicability of the findings. Moreover, the edge-rewiring experiments provide compelling evidence linking community structure to path multiplicity. The proposed TSF generative model successfully captures the observed empirical behavior, outperforming standard random-graph models and opening avenues for further theoretical work. Overall, these contributions advance our understanding of structural redundancy and resilience in complex networks.

On the other hand, it is increasingly recognized that purely structural metrics often have limited explanatory power without a clear link to network functions or dynamics (Gates & Rocha, 2016 for network control; Hines et al., 2010; Pahwa et al., 2012 for power grids; Zhang et al., 2015 for transportation systems). Therefore, introducing a new structural metric — especially one limited to unweighted networks — requires a strong justification of its utility beyond existing descriptors. To strengthen the paper’s impact, the authors are encouraged to further elaborate on the practical importance of path multiplicity. This could be achieved by discussing or demonstrating its role in a functional or dynamic context, such as network routing, spreading processes, or resilience to failures, showing how the metric offers truly novel insights not captured by classical measures.

Response: We sincerely thank the reviewer for this positive overall assessment and insightful comment on the practical importance of path multiplicity. As a structural metric, path multiplicity is linked to several key aspects of network function and dynamics. First, it serves as a structural proxy for robustness and reliability of connectivity, since multiple shortest paths can both buffer local failures and concentrate load on shared bottlenecks, shaping vulnerability to congestion and cascades. It also modulates transport, diffusion, and spreading by providing multiple equally short channels that accelerate propagation and redistribute flows or probabilities across routes. Finally, by increasing ambiguity in path choice and revealing node pairs connected through rich internal structure, path multiplicity is likely to affect navigability, decision-making, and influence or coordination in multi-agent and control processes.

In the revised version, we have added a description of the practical importance of path multiplicity to the third paragraph of the *Introduction* as follows.

Unlike classical metrics that focus solely on single optimal paths, path multiplicity adds a new layer of complexity when attempting to understand network structure. As a structural metric, path multiplicity is linked to key aspects of network function and dynamics. It serves as a structural proxy for robustness and reliability of connectivity, since multiple shortest paths can both buffer local failures and concentrate load on shared bottlenecks, shaping vulnerability to congestion and cascades. It also modulates transport, diffusion, and spreading by providing multiple equally short channels that accelerate propagation and redistribute flows or probabilities across routes. Finally, by increasing ambiguity in path choice and revealing node pairs connected through rich internal structure, path multiplicity is likely to affect navigability, decision-making, and influence or coordination in multi-agent and control processes. However, the mechanisms behind path multiplicity remain largely unexplored. What mechanisms lead to the hesitant-world phenomenon? What mechanisms drive the power-law distribution in path multiplicity? In this study, we introduce the concept of relative path multiplicity, which allows us to systematically compare the path multiplicity of real-world networks to that of equivalent random networks. With this approach, we explore the underlying structural factors that contribute to the path multiplicity in complex networks. Furthermore, we develop a new network model to replicate the observed characteristics in empirical networks, offering insights into the mechanisms underlying the hesitant-world phenomenon and the power-law distribution of path multiplicity.

Comment 1: A discussion/analysis on how the conclusions might change for weighted and/or directed edges would be important.

Response 1: Thank you for your helpful comments. We agree that the conclusions might change for weighted and/or directed edges. Following your suggestion, we have added descriptions in the third paragraph of the third paragraph of *Discussion* as follows.

Notably, this paper restricts attention to undirected and unweighted networks. Considering that edge directionality and weights can substantially reshape path multiplicity—altering tie patterns and thereby inflating or suppressing the number of shortest paths—the phenomena and mechanisms of path multiplicity in directed and weighted networks remain to be studied in the future. Moreover, this paper draws evidence from empirical networks and simulations; a fuller theoretical account remains to be developed. Promising high-impact research directions could include an analytical relationship between path multiplicity and structural parameters, spectral approaches to bound the path multiplicity, and cut/flow arguments via Menger’s theorem that link cross-community cut size and boundary redundancy.

Comment 2: I would be hesitant using the term “path hesitation” to convey the concept of path multiplicity.

Response 2: We appreciate the concern about the term “path hesitation”. In our prior article (Ye Deng et.al., PNAS Nexus, 2024), we have employed the terms “path hesitation amount (PHA)”, “path hesitation matrix (PHM)”, “path hesitation index (PHI)” to capture the intuition that, in networks with a large number of shortest paths between nodes, routing or navigation decisions become more complex and may induce hesitation in the decision maker. To maintain consistency in terminology, we retain this legacy term. We will continue to reflect on the terminology and will adopt a more standard wording in future work if a clearer community consensus emerges.

Comment 3: The causal attribution of RPHI to community structure is dependent on the specific community detection algorithm used. The authors should specify which algorithm and resolution parameter were employed.

Response 3: Thank you for your helpful comments. We agree that the causal attribution of RPHI to community structure is dependent on the specific community detection algorithm used. In this study, we employ Newman’s leading-eigenvector method for community detection. We have also tested other algorithms, such as Louvain, Leiden, and Infomap, and obtained

qualitatively similar results. Following your suggestions, we have added statements in *Sec. 2.1* and the specific algorithm in the *Methods* as follows.

Sec. 2.1 Concept of relative path multiplicity

The PHI reflects the complexity of path selection within the network: a large PHI suggests that the network has a hesitant-world property, with many shortest paths between node pairs, potentially leading to a more complex decision-making process. To reveal the relationships between the PHI and classical network metrics, we present corresponding scatter plots along with the Pearson's correlation coefficient ρ in 140 real-world networks in Fig.1a. The community detection algorithm used in this study is Newman's deterministic leading-eigenvector modularity algorithm (Newman, 2006); the specific algorithmic steps and parameter settings are detailed in the Methods section, and we remark that our main conclusions are robust to the choice of community detection algorithm. For a more in-depth exploration, we report the quadrant consistency rate (QCR) (Choudhary et al., 2017) as a robustness check alongside conventional correlations and use it to probe the dependencies that ordinary coefficients may miss. The QCR measures the fraction of observations whose deviations from their respective medians share the same sign and thus captures concordance in a robust, rank-like manner (see the Methods section for details). A significantly higher proportion of data points within the consistent quadrants in the quadrant analysis would indicate a stronger correlation between the variables. The scattered points are clearly widely dispersed throughout the plane and the values of ρ and QCR are quite low, suggesting that there are no significant linear correlation between the PHI and the classical network metrics.

Methods

Community detection via leading-eigenvector modularity

Non-overlapping communities are detected by maximizing the modularity of the undirected graph $G (V, E)$ in a deterministic spectral framework. Let $A(G) = (a_{ij})_{N \times N}$ denote the adjacency matrix defined above, and $2M = \sum_{i,j} a_{ij}$. Define the modularity matrix

$$B = A(G) - \frac{\mathbf{k}\mathbf{k}^T}{2M}$$

where $\mathbf{k} = (k_1, \dots, k_N)^T$

For a bipartition encoded by $\mathbf{s} \in (\pm 1)^N$, the modularity can be written as

$$Q = \frac{1}{4M} \mathbf{s}^T B \mathbf{s}$$

Let \mathbf{u}_1 be the leading eigenvector of B . An initial split is obtained by the sign rule $s_i = \text{sign}(u_{1i})$. A single-vertex refinement is then applied: iteratively move the node that yields the largest positive modularity gain ΔQ , and repeat until no further improvement is possible $\Delta Q \leq 0$.

To obtain more than two groups, recursion is applied to any accepted vertex subset $U \subseteq V$ (a current community). The net modularity change from subdividing U is evaluated with the generalized modularity matrix

$$B_{ij}^{(U)} = B_{ij} - \delta_{ij} \sum_{\ell \in U} B_{i\ell}, \quad i, j \in U$$

and the same spectral-plus-refinement steps are applied within U . A subdivision is accepted only if it increases Q ($\Delta Q > 0$); recursion stops when all current groups are indivisible under this criterion.

Matrix–vector products with B are evaluated as

$$B\mathbf{x} = A(G)\mathbf{x} - \frac{\mathbf{k}^T \mathbf{x}}{2M} \mathbf{k}$$

so each multiply costs $O(M+N)$. Using the power or Lanczos method for \mathbf{u}_1 , the overall cost is typically $O(N^2)$ on sparse graphs. The procedure is deterministic given $A(G)$ (up to a global sign flip of \mathbf{u}_1 , which does not affect the partition). The algorithm returns the final community labels $\{c_i\}_{i=1}^N$ and the achieved modularity Q . In practice, we used the Brain Connectivity Toolbox function *modularity_und* (Rubinov et al., 2010) with the resolution parameter fixed at its default value $\gamma = 1$ for all community detection analyses.

Comment 4: The quadrant-consistency rate (QCR) is central to the analysis but does not appear to be a standard metric in network science. The authors should provide a justification for using QCR.

Response 4: Thank you for your helpful comments. QCR measures the fraction of observations whose deviations from their respective medians share the same sign and thus captures concordance in a robust, rank-like manner. Its value lies in complementing standard coefficients such as Pearson’s correlation coefficient and Spearman’s correlation coefficient: when associations are monotonic but nonlinear, masked by heavy tails or outliers, or split across clusters that depress the correlation coefficient, QCR can still reveal systematic alignment between variables. For this reason, we report QCR as a robustness check alongside conventional correlations, using it to probe dependencies that ordinary coefficients may miss.

In the revised version, we have added descriptions in *Sec. 2.1* and *Methods* as follows.

Sec. 2.1 Concept of relative path multiplicity

The PHI reflects the complexity of path selection within the network: a large PHI suggests that the network has a hesitant-world property, with many shortest paths between node pairs, potentially leading to a more complex decision-making process. To reveal the relationships between the PHI and classical network metrics, we present corresponding scatter plots along with the Pearson's correlation coefficient ρ in 140 real-world networks in Fig1.a. The community detection algorithm used in this study is Newman's deterministic leading-eigenvector modularity algorithm (Newman, 2006); the specific algorithmic steps and parameter settings are detailed in the Methods section, and we remark that our main conclusions are robust to the choice of community detection algorithm. For a more in-depth exploration, we report the quadrant consistency rate (QCR) (Choudhary et al., 2017) as a robustness check alongside conventional correlations and use it to probe the dependencies that ordinary coefficients may miss. The QCR measures the fraction of observations whose deviations from their respective medians share the same sign and thus captures concordance in a robust, rank-like manner (see the Methods section for details). A significantly higher proportion of data points within the consistent quadrants in the quadrant analysis would indicate a stronger correlation between the variables. The scattered points are clearly widely dispersed throughout the plane and the values of ρ and QCR are quite low, suggesting that there are no significant linear correlation between the PHI and the classical network metrics.

Methods

Quadrant consistency rate

To quantify the monotonic relationship between relative path hesitation and a given network metric, we employ the quadrant consistency rate (QCR), a non-parametric measure of association derived from quadrant analysis. The method partitions the scatter plot of two variables into four quadrants using robust reference lines (e.g., medians) and evaluates the proportion of points whose deviations from these reference lines are directionally consistent.

Given paired observations $\{(x_i, y_i)\}_{i=1}^N$ for two variables X and Y , define \tilde{x} and \tilde{y} as the sample medians of $\{x_i\}$ and $\{y_i\}$, respectively. The plane is divided by the vertical line $x = \tilde{x}$ and the horizontal line $y = \tilde{y}$ into four quadrants:

Quadrant I ($x \geq \tilde{x}, y \geq \tilde{y},$);

Quadrant II ($x < \tilde{x}, y \geq \tilde{y},$);

Quadrant III ($x < \tilde{x}, y < \tilde{y},$);

Quadrant IV ($x \geq \tilde{x}, y < \tilde{y}$).

We count the number of points $n_I, n_{II}, n_{III}, n_{IV}$ that fall in each quadrant. If a positive monotonic relationship is hypothesized, points in Quadrant I and Quadrant III share the same sign of deviation from the medians and are deemed consistent, whereas points in Quadrants II and Quadrant IV are inconsistent. Thus, the QCR is defined as

$$QCR_+ = \frac{n_I + n_{III}}{N},$$

which takes values in $[0,1]$. A value close to 1 indicates that most observations lie in the first or third quadrant, implying a strong positive association, while a value near 0.5 suggests no directional preference. When a negative monotonic association is hypothesized, we instead treat Quadrants II and IV as consistent and define

$$QCR_- = \frac{n_{II} + n_{IV}}{N},$$

In both cases, the QCR measures the proportion of data points that exhibit concordant deviations and is robust to outliers due to the use of medians.

Comment 5: Figure 1 is unfortunately confusing for an introductory figure. For example, while its caption promises to show a “Relationship between the path multiplicity and classical network metrics,” the data and the demonstration do not reveal any relationship.

Response 5: Thank you for your careful comments. The goal of Fig. 1 is to explore potential factors influencing path multiplicity. In Fig. 1a, we present scatter plots for PHI and classical network metrics in 140 real-world networks. The results indicate no significant linear correlation between PHI and the classical network metrics. Considering that the edge density may affect path multiplicity, we display the PHI as a function of edge density p in Figs. 1b and 1c, which motivates our focus on the relative path hesitation index (RPHI).

Following your comment, we have modified the caption of Fig. 1 as “*Potential factors influencing path multiplicity*”. Also, we have refined descriptions of the motivation for Fig. 1 in *Sec. 2.1* as follows.

Intuitively, according to the definition of the PHI, the edge density may affect path multiplicity. The simple example network shown in Fig. 1b is initially a chain network with an initial PHI value of 1. As the edge density increases, and thus the network becomes fully connected, the corresponding PHI value returns to 1. Furthermore, the changing trend of PHI as the edge density p increases is shown in Fig. 1c for three typical model networks: Erdős-Rényi random networks (ER), regular ring lattices (RRL), and cluster Barabási-Albert scale-free networks (CBA). Unsurprisingly, one can see that the PHI values are strongly related to the edge density p with some form of complex nonlinear relationship. To analyze the intrinsic factors influencing the PHI, it is necessary to exclude the effects of network size and edge density on path multiplicity. Thus, it inspires us to introduce the concept of the relative path hesitation index (RPHI), denoted by $\tilde{\Phi}(G)$, which is calculated by normalizing the PHI of the network G with the PHI of an equivalent ER random network G_{ER} with the same size and edge density as follows:

$$\tilde{\Phi}(G) = \frac{\Phi(G)}{\Phi(G_{ER})}$$

Comment 6: Typos: “Path multiplicity <has> profound implications for network efficiency and route decision-making” – on page 2 and some others.

Response 6: Thank you for flagging these typos. We have corrected the specific instance and performed a thorough proofread across the manuscript. In addition, we engaged a professional English-language editing service to improve grammar, style, and consistency throughout.

Comment 7: Could a high RPHI also arise from a core-periphery structure? For example, in a network with few communities but a very densely connected core.

Response 7: Thank you for your insightful comments. Following your suggestion, we have conducted experiments to test whether high path multiplicity can arise from a core-periphery structure. We show in Fig. R1 the PHI values of 32 real-world networks along with the PHI values of reference model networks with the same network size and similar edge density, including Erdős-Rényi (ER) random networks, Newman-Watts (NW) small-world networks, Barabási-Albert (BA) scale-free networks, cluster Barabási-Albert (CBA) scale-free networks, and core-periphery structure networks. We find that, like other classical model networks, core-periphery networks (Fig. R1e) also could not yield high RPHI values.

Fig. R1. Scatter plots for $\Phi(G)$ of 32 real networks and corresponding model networks with the same network size and edge density. The dashed line and the darker shaded area denote a reference slope of 1.

Holland, Paul W., Kathryn Blackmond Laskey, and Samuel Leinhardt. "Stochastic blockmodels: First steps." *Social networks* 5.2 (1983): 109-137.

Borgatti, Stephen P., and Martin G. Everett. "Models of core/periphery structures." *Social networks* 21.4 (2000): 375-395.

Comment 8: The TSF model assumes a single power-law exponent, λ , across all communities. Given that real-world networks often exhibit community-specific exponents (Nature 435, 814-818, 2005, Phys. Rev. E 94, 012302, 2016), a discussion on the implications of this simplifying assumption and potential model extensions would be welcome.

Response 8: Thank you for your helpful comments. We agree that, in real-world networks, scaling exponents can differ across communities. Following your suggestion, we have modified the model description in *Methods* and added references as follows. In the modified model, power-law exponent λ can differ across communities.

Tribal-Scale-Free (TSF) Network Model

The Tribal Scale-Free (TSF) model is a generative model designed to capture the hierarchical and modular features typical of real-world networks, adding the separation of distinct communities (cf. the “connected caveman model”). The TSF model builds a network by integrating intracommunity scale-free subnetworks with controlled intercommunity connections, allowing for an examination of the relationship between modularity and path multiplicity. The detailed mathematical formulation of the TSF model is presented below, with a step-by-step description of its generation process.

Step 1: Initialization. Start with a total network size N and a specified number of communities n_c . The N nodes are randomly assigned to n_c distinct communities, representing different “tribes”. Let C_i denote the i -th community, where $i \in 1, 2, \dots, n_c$. The size of each community $|C_i|$ is determined such that

$$\sum_{i=1}^{n_c} |C_i| = N$$

Step 2: Generation of scale-free subnets. Within each community C_i , a scale-free network is generated by using the configuration model (CM). The CM constructs a random graph with a prescribed degree distribution, ensuring that the degree distribution follows a power-law characterized by an exponent λ_i , while maintaining the fixed average degree $\langle k_i \rangle$ within each community. The degree distribution adheres to the following power-law form:

$$P(k) \sim k^{-\lambda_i}, \text{ for } k \geq k_{min}.$$

In this step, the network degree sequence is predefined, and the model ensures that the specified degree distribution is achieved by appropriately matching the node degree constraints.

Step 3: Intercommunity edge allocation. Once scale-free intracommunity subnets are generated, proceed with intercommunity edge allocation. Given a predefined number of intercommunity edges n_e , two distinct communities, C_i and C_j , are selected, and then n_e edges are created between them. For subsequent connections, an unconnected community C_m is chosen uniformly at random, and this community is linked with n_e edges to any of the already connected communities. This process is repeated iteratively until all of the communities are connected, ensuring the formation of a single connected component throughout the network.

Step 4: Construction and finalization of the network. The TSF network $G = (V, E)$ consists of $|V| = N$ nodes and a set of edges E , combining the intracommunity edges E_{intra} and the intercommunity edges E_{inter} such that $E = E_{intra} \cup E_{inter}$. The total number of edges in the network is expressed as follows:

$$|E| = \sum_{i=1}^{n_c} |E_{intra}^i| + n_e \cdot (n_c - 1).$$

Notably, to keep the model simple, the proposed TSF model assumes a common scaling exponent across communities. In practice, scaling exponents can differ between communities in real-world networks (Palla et al., 2005; Stegehuis et al., 2016;), and future extensions of the TSF model can therefore allow heterogeneous, community-specific scaling exponents rather than enforcing a single global value.

Furthermore, the TSF model can effectively capture the modular and hierarchical nature of complex networks by combining scale-free substructures within communities and interconnecting them through a controlled number of intercommunity edges. By adjusting the parameters n_c (number of communities) and n_e (number of intercommunity edges), the model facilitates a systematic exploration of the impact of community structure on network properties such as the path multiplicity and overall network efficiency.

Palla, G., Derényi, I., Farkas, I., & Vicsek, T. (2005). Uncovering the overlapping community structure of complex networks in nature and society. *nature*, 435(7043), 814-818.

Hocine C, Palla, Boleslaw K. Szymanski, Xiaoyan Lu. On community structure in complex networks: challenges and opportunities. *Applied Network Science* 4.1 (2019): 1-35.

Stegehuis, C., Van Der Hofstad, R., & Van Leeuwen, J. S. (2016). Power-law relations in random networks with communities. *Physical Review E*, 94(1), 012302.

Comment 9: The rewiring method in Sec. 4.2 preserves the degree sequence and global connectivity but does not constrain other topological properties like clustering, diameter, or assortativity. Since these metrics could also influence PHI, it would be helpful to report their evolution during the optimization process.

Response 9: Thank you for your helpful comments. Following your suggestions, we have added experiments to report the evolution of clustering, diameter, and assortativity during the optimization process. We find that there is no significant causal relationship between these topological properties and path multiplicity. For example, in the ER random network shown in the first subplot of Fig. 2R(a), as the value of $\Phi(G)$ monotonically increases from 3.90 to 21.65, the assortativity coefficient r first decreases and then increases. Moreover, in BA scale-free networks, r remains nearly constant as $\Phi(G)$ increases.

In the revised version, we have placed the figures in the Appendix to keep the main text compact.

a. Variation of r values with incremental PHI optimization

b. Variation of PHI values with incremental r optimization

Fig. R2. Target-oriented edge rewiring in three types of model networks. a, Variation in the assortativity r values with incremental optimization of the PHI through target-oriented edge rewiring. As the PHI increases, the value of assortativity r exhibits a model-dependent, nonmonotonic trend. b, Variation in the PHI values with incremental optimization of assortativity r through target-oriented edge rewiring. In this scenario, edge rewiring is directed toward increasing r , resulting in a nonmonotonic (often U-shaped) trend in the PHI values.

Fig. R3. Target-oriented edge rewiring in three types of model networks. a, Variation in the clustering coefficient C values with incremental optimization of the PHI through target-oriented edge rewiring. As the PHI increases, the value of the clustering coefficient C shows a weak, model-dependent trend consisting of small increases in ER/NW networks and near constancy in BA networks. b, Variation in the PHI values with incremental optimization of the clustering coefficient C through target-oriented edge rewiring. In this scenario, edge rewiring is directed toward increasing C , resulting in a nonmonotonic (often U-shaped) trend in the PHI values.

Fig. R4. Target-oriented edge rewiring in three types of model networks. a, Variation in the diameter D values with incremental optimization of the PHI through target-oriented edge rewiring. As the PHI increases, the value of diameter D generally increases with the model-dependent fluctuations. b, Variation in the PHI values with incremental optimization of the diameter D through target-oriented edge rewiring. In this scenario, edge rewiring is directed towards increasing D , resulting in a highly nonmonotonic, spike-like trend with intermittent peaks and no sustained increase in the PHI values.

Comment 10: Perhaps the paper's theoretical contribution could also be strengthened. Have the authors considered deriving an analytical relationship between path multiplicity and community structure parameters, for instance within a stochastic block model? Another valuable approach would be to use spectral methods to obtain rigorous theoretical bounds on path multiplicity.

Response 10: Thank you for your insightful suggestions. We have now been trying to explore path multiplicity theoretically. For example, for the elementary Erdős–Rényi (ER) random graph, we have derived a closed-form characterization of path multiplicity as $\Phi(G) = p + N(1 - p)p^2$ and proved that the path multiplicity reaches its maximum values at $p = 2/3$. We have actively pursued analytical extensions beyond ER random networks, but principal

obstacles remain. In more sophisticated network models, such as the stochastic block model, shortest paths overlap around bridges/hubs and thus lose independence; the number of length-equivalent configurations grows combinatorially; and modular boundary conditions (intra-/inter-block) break the symmetries. Thus, deriving an analytical relationship between path multiplicity and community-structure parameters remains a challenging open problem and appears to require additional theoretical tools.

We strongly agree that it is a valuable approach to obtain rigorous theoretical bounds on path multiplicity using spectral methods. Considering that this paper primarily focuses on the impact of community structure on path multiplicity, we will separately explore the upper and lower bounds on path multiplicity in future work.

In the revised version, we have provided an outlook on future work in the third paragraph of *Discussion* as follows.

Notably, this paper restricts attention to undirected and unweighted networks. Considering that edge directionality and weights can substantially reshape path multiplicity—altering tie patterns and thereby inflating or suppressing the number of shortest paths—the phenomena and mechanisms of path multiplicity in directed and weighted networks remain to be studied in the future. Moreover, this paper draws evidence from empirical networks and simulations; a fuller theoretical account remains to be developed. Promising high-impact research directions could include an analytical relationship between path multiplicity and structural parameters, spectral approaches to bound the path multiplicity, and cut/flow arguments via Menger's theorem that link cross-community cut size and boundary redundancy.

✧ **Reviewer #3:**

Comment 1: I co-reviewed this manuscript with one of the reviewers who provided the listed reports. This is part of the Nature Communications initiative to facilitate training in peer review and to provide appropriate recognition for Early Career Researchers who co-review manuscripts.

Response 1: Thank you for your insightful comments. We have provided a point-by-point response to every comment in the reports and indicated the corresponding revisions with page/line references; all changes are highlighted in the manuscript.

Point-by-point Responses to Reviewers' Comments

✧ **Reviewer #1:**

General assessment: Thank you for updating your manuscript following my suggestions.

Response: We sincerely appreciate your time and effort in reviewing our manuscript.

✧ **Reviewer #2:**

General assessment: I thank the authors for considering my previous comments and providing detailed feedback. While the manuscript has certainly improved, I am afraid that the authors have not fully addressed the main concerns. Again, the concept of path multiplicity is definitely interesting. However, I do not think the overall quality and rigor of the manuscript, even after the revision, meet the standards for Nature Communications. I have the following lingering concerns. Overall, I still believe the paper will be of interest to specific readers (myself included). However, the current quality of the presentation is concerning. I hope the authors take these comments seriously and undertake a more substantial revision.

Response: Thank you very much for your careful assessment and for highlighting the remaining concerns. We are sorry that our previous revision did not fully address the key issues you raised. We have taken your comments very seriously and undertake a substantially deeper and more rigorous revision to improve both the quality and clarity of the manuscript in this revision.

Comment 1: Previously, I noted that introducing a new structural metric, particularly one restricted to unweighted networks, requires strong justification. In response, the authors provided a discussion on the relevance of path multiplicity, which feels brief and generic. No references are cited to support this discussion. No justification of the necessity (or unnecessary) of considering edge weights is given. The meanings of “structural proxy” and “increasing ambiguity” remain vague in this context. The transition to the statement that “...the mechanisms behind path multiplicity remain largely unexplored...” also feels abrupt and disrupts the flow.

Response 1: Thank you for your insightful and constructive feedback, which has helped us to significantly strengthen the motivation and clarity of our work. We have revised the manuscript to provide a more substantial discussion on the functional importance of path multiplicity and to clarify the scope and logic of our study. We hope these revisions root the study of path multiplicity within the broader network science discourse. Thank you for guiding us toward these improvements.

1. Justification of the practical importance of path multiplicity

We have expanded the discussion to explicitly link path multiplicity to well-established themes in network science, supported by specific citations. We now detail its role in:

- **Robustness and Congestion:** How multiple shortest paths create redundancy but also potential shared bottlenecks (Sole-Ribalta et al., *New J. Phys.*, 2019).
- **Diffusion and Transport:** How they provide parallel channels that can accelerate spreading and redistribute flows (Gómez et al., *Phys. Rev. Lett.*, 2013).
- **Routing and Decision-Making:** How they introduce navigational choice and potential decision paralysis (Chernev et al., *J. Consumer Psychol.*, 2015; Kleinberg, *Nature*, 2000).

The second paragraph and third paragraph of the *Introduction* is revised as follows:

Despite the advances in the study of paths, path multiplicity—the number of equidistant shortest paths between node pairs—remains unexplored. Recently, some studies have revealed the existence of a universal power-law scaling of path multiplicity, indicating that a few node pairs have a large number of shortest paths (Deng et al., *PNAS Nexus*, 228, 2024). For example, even in the Bn-Macaque-Rhesus-Brain-1 brain network with only 242 nodes (Rossi et al., *AAAI*, 29, 2015), the largest count of shortest paths between node pairs can reach 649, with an average value of 11.07. This observation reveals that the world we live in is not just a “small-world”, but also a “hesitant-world”. The “small-world” effect means that most nodes are connected through only a few steps, even in a large network; meanwhile, the “hesitant-world” effect implies that one may hesitate among numerous choices, even when the network itself is small.

Although the multiplicity of shortest paths is a structural feature, it actively governs how networks function. First, path multiplicity shapes a network's robustness and vulnerability. While redundant shortest paths can preserve connectivity when links fail, they may also concentrate traffic on common bridges or hubs, creating bottlenecks that increase susceptibility to congestion or targeted attacks—a dual role noted in studies of network flows (Sole-Ribalta et al., *New J. Phys.* 21, 2019). Second, it affects spreading processes and transport. Multiple, equally short routes can accelerate the propagation of information, diseases, or resources by offering parallel transmission channels, while also redistributing loads across the network and influencing diffusion outcomes (Gómez et al., *Phys. Rev. Lett.* 110, 2013). Finally, it has consequences for routing and decision-making. For any agent—whether a packet, an individual, or an algorithm—navigating among many “optimal” paths can induce “choice overload,” a psychological effect known to impair decision efficiency (Chernev et al., *J. Consumer Psychol.* 25, 2015), potentially degrading the performance of navigation strategies (Kleinberg, *Nature.* 406, 2000).

2. Justification of the necessity of considering edge weights

We agree that this is a critical direction for extending the understanding of path multiplicity. We have carefully considered your point and have refined our discussion to focus specifically on how weights influence path multiplicity and to outline meaningful next steps for research in weighted networks. We have studied in particular the following:

- **Impact of Edge Weights:** In weighted networks, the definition of the "shortest path" shifts from a topological distance to a sum of edge weights. This dramatically reduces the number of qualifying paths. Consequently, the extreme multiplicity observed in unweighted topologies is likely suppressed in many weighted scenarios, often resulting in a single, unique shortest path.

- **Current Knowledge Gap:** This highlights a significant gap in our understanding. Since most real-world networks are inherently weighted, the multiplicity of near-optimal or functionally equivalent routes becomes the more relevant operational concept. The current theory of path multiplicity cannot directly capture this.
- **Constructive Future Direction:** A promising avenue is to generalize this concept. One could define a "tolerance parameter" (ϵ) that allows paths with total weight within $(1+\epsilon)$ of the absolute minimum to be considered as "tolerant shortest paths." Analyzing the multiplicity of these ϵ -tolerance paths could bridge our structural insights with weighted reality, revealing how network architecture provides robustness and alternative routing options under realistic, non-exact conditions.

The second paragraph of the *Discussion* is revised as follows:

This study identifies community structure as the key driver of path multiplicity in unweighted networks. A crucial next step is to extend this insight to weighted networks—the mode of most real-world systems, from transportation to social interactions. Edge weights fundamentally alter the shortest path landscape: the condition for multiplicity (multiple paths with identical total weight) becomes mathematically strict, often reducing many shortest paths to a single optimal route. This shift reveals an important gap and opportunity, as practical applications often concern near-optimal alternatives rather than perfectly equivalent paths. To bridge this gap, future work could generalize the concept by introducing such a tolerance parameter $\epsilon \geq 0$, defining the paths within $(1+\epsilon)$ of the minimum weight as “tolerant shortest paths.” Analyzing the multiplicity of such paths would clarify how network architecture—particularly the community structure—supplies robust routing options under realistic conditions, directly linking our topological findings to challenges in resilient infrastructure, adaptive traffic routing, and the analysis of neural or social networks with continuous link strengths.

3. **Refinement of the Introduction:** We have reorganized the *Introduction* to make it clearer and accessible. The logical flow now progresses from (a) the empirical discovery of a power-law in path multiplicity, to (b) its recognized functional implications, to (c) the outstanding question of its structural origin, which our study directly addresses.

The third paragraph of the *Introduction* is revised as follows:

Our recent finding (Deng et al., *PNAS Nexus*, 228, 2024)—a strong power-law distribution in path multiplicity across diverse real-world networks—reveals that the path multiplicity is a fundamental and widespread architectural signature. Given the functional relevance outlined above, a central unanswered question emerges: which underlying network structure produces such extreme heterogeneity in path multiplicity? Uncovering this structural origin is essential for explaining observed network dynamics and for designing networks with desired functional properties. In this

study, we introduce the concept of *relative path multiplicity* to systematically compare real-world networks with equivalent random networks. Using this approach, we identify the structural factors that shape path multiplicity in complex networks. Furthermore, we develop a new network model that can reproduce the empirical characteristics, offering mechanistic insights into the observed power-law distribution of path multiplicity.

Comment 2: I also noted that the correlation between RPMI and community structure is likely dependent on the specific community detection algorithm used, which the authors agreed: "...We agree that the causal attribution of RPMI to community structure is dependent on the specific community detection algorithm used..." However, this specification or limitation is not mentioned in the manuscript, which risks exaggerating the paper's conclusion. While the authors claim that "...main conclusions are robust to the choice of community detection algorithm..." and stated in their reply that they tested algorithms such as Louvain, Leiden, and Infomap, no results are given in the Main Text, Methods, or Supplementary Information. Given the authors' statement that "...this paper primarily focuses on the impact of community structure on path multiplicity," it seems biased to report results for only a single algorithm. I urge the authors to include these comparative results to substantiate their claim.

Response 2: We sincerely thank you for this helpful comment. We agree that any empirical correlation between RPMI and community structure can depend on how communities are operationalized, i.e., the specific community detection algorithm used. In our initial revision, we did not make this limitation sufficiently explicit in the manuscript text, which could indeed leave room for overstatement. To address this concern, we have now performed a comprehensive analysis based on various community detection methods and reported the corresponding comparative results in the *Supplementary Information*.

Specifically, we performed the correlation analysis between RPMI and community structure using six different algorithms, including Leading Eigenvector (*Newman, M.E et al, 2006*), Walktrap (*Pons, P et al, 2006*), Leiden (*Traag, V.A et al, 2019*), Label Propagation (*Raghavan, U.N et al, 2007*), Infomap (*Rosvall, M et al, 2008*), and Louvain (*Blondel, V.D et al, 2008*). The resulting correlation plots are now provided in the *Supplementary Information*. All community detection packages/functions and parameter settings used in this study, together with a brief documentation, are available at <https://github.com/gituserbnu/path-multiplicity.git>. In addition, we have revised the Main Text to explicitly state that our main conclusions are robust to the choice of community detection algorithm.

In the revised version, we have added descriptions in *Sec. 2.2* and *Supplementary Information* as follows.

Sec. 2.2 Intrinsic Factors of Path Multiplicity

Considering that a specific community detection algorithm may influence the causal attribution of RPMI to community structure, we have tested classical community detection algorithms, including the Leading Eigenvector method (Newman et al., *Phys. Rev. E.* 74(3), 036104, 2006), Walktrap method (Pons et al., *J. Graph Algorithms Appl.* 10(2), 191–218, 2006), Leiden method (Traag et al., *Sci. Rep.* 9(1), 1–12, 2019), Label Propagation method (Raghavan et al., *Phys. Rev. E.* 76(3), 036106, 2007), Infomap method (Rosvall et al., *Proc. Natl. Acad. Sci.* 105(4), 1118–1123, 2008), and Louvain method (Blondel et al., *J. Stat. Mech.: Theory Exp.* 2008(10), 10008, 2008). Our experimental results are reported in the Supplementary Information. Although these algorithms yield somewhat different community counts, the Spearman correlation between RPMI and the number of communities remains consistently high (all $\rho_s > 0.8$), and all the corresponding QCR values are above 0.9. These results indicate that our findings do not depend on any particular community detection algorithm; accordingly, we remark that our main conclusions are robust to the choice of community detection algorithm.

Sec. Supporting Information

Table R1. Community counts of 140 real-world networks under multiple community detection algorithms. Columns correspond to the number of communities detected by each algorithm.

No	Name	N	Leading Eigenvector	Walktrap	Leiden	Label Propagation	Infomap	Louvain
1	bn-cat-mixed-species_brain_1	65	3	4	4	1	1	4
2	bn-macaquerhesus_brain_2	91	2	1	3	1	1	4
3	bn-macaquerhesus_cerebralcortex_1	91	3	1	3	1	1	3
4	bn-macaquerhesus_interareal-cortical-network_2	93	1	1	1	1	1	1
5	bn-mousekasthuri_graph_v4	987	21	117	20	82	109	22
6	bn-mouse_brain_1	213	2	2	3	1	1	3
7	bn-mouse_retina_1	1076	4	10	5	1	1	5
8	bn-mouse_visualcortex_1-2	29	3	6	3	3	4	5

9	aves-barn-swallow-contact-network	17	2	4	3	1	1	2
10	aves-barn-swallow-non-physical	17	2	2	2	1	1	2
11	aves-sparrow-social	52	4	5	4	1	1	4
12	aves-sparrow-social-2009	31	3	4	3	3	2	3
13	aves-sparrow-social-2010	40	4	3	3	1	1	3
14	aves-sparrowlyon-flock-season2	43	3	7	3	1	1	3
15	aves-sparrowlyon-flock-season3	27	2	2	2	1	1	2
16	aves-thornbill-farine	63	3	1	3	1	1	3
17	bio-CE-GT	878	22	61	17	29	51	17
18	bio-CE-LC	993	91	59	27	95	104	27
19	bio-CE-PG	734	3	1	4	2	1	4
20	bio-DM-HT	2831	243	336	45	292	310	43
21	bio-DM-LC	483	34	58	18	47	52	18
22	bio-DR-CX	3287	4	170	7	2	143	5
23	bio-HS-HT	2499	50	293	20	49	204	19
24	bio-SC-LC	1999	15	276	10	6	106	10
25	bio-SC-TS	101	2	1	2	1	1	3
26	bio-celegans	453	8	35	9	6	36	10
27	bio-celegans-dir	453	8	35	10	6	37	11
28	bio-grid-mouse	791	66	92	26	76	84	26
29	bio-grid-plant	1272	137	104	28	97	102	27
30	bio-grid-worm	3343	36	269	27	103	277	31
31	bio-yeast	1458	152	174	32	172	175	31
32	bio-yeast-protein-inter	1458	152	174	32	174	180	32
33	ca-AstroPh	17903	611	2264	38	347	898	37
34	ca-CSphd	1025	138	66	33	136	137	32
35	ca-CondMat	21363	979	2306	56	1552	1337	54
36	ca-Erdos992	4991	430	197	36	330	296	33
37	ca-GrQc	4158	494	448	42	375	367	40
38	ca-HepPh	11204	206	2631	43	368	758	39
39	eco-everglades	69	3	2	3	1	1	4
40	eco-foodweb-baydry	128	4	2	4	1	1	4
41	eco-foodweb-baywet	128	4	2	4	1	1	5
42	eco-mangwet	97	3	2	4	1	1	4
43	eco-stmarks	54	4	8	4	1	1	4
44	ENZYMES_g118	95	11	13	9	16	14	9
45	ENZYMES_g196	50	6	5	5	9	7	5
46	ENZYMES_g198	55	6	9	8	7	10	6

47	ENZYMES_g2 03	56	5	6	5	7	9	5
48	ENZYMES_g2 09	57	6	6	6	6	10	6
49	ENZYMES_g3 49	64	7	6	6	8	9	6
50	ENZYMES_g4 65	52	7	8	7	9	9	7
51	ENZYMES_g4 84	60	5	9	7	4	10	7
52	ENZYMES_g5 01	66	8	8	9	16	11	9
53	ENZYMES_g5 32	74	7	12	6	13	12	5
54	ENZYMES_g5 41	28	6	5	5	5	5	5
55	econ-mahindas	1258	10	195	8	3	120	9
56	econ-poli	2343	216	88	45	216	208	45
57	econ-wm3	257	5	8	6	4	19	4
58	ia-email-univ	1133	14	49	11	15	77	11
59	ia-hospital- ward-proximity	75	3	3	4	1	1	4
60	ia-hospital- ward-proximity- attr	75	3	3	4	1	1	5
61	ia-infect-hyper	113	4	5	5	1	1	6
62	ia-reality	6809	337	68	45	81	91	43
63	ia- southernwomen	18	2	2	2	2	1	2
64	ia-workplace- contacts	92	4	4	4	1	4	4
65	power-1138-bus	1138	56	87	27	133	134	25
66	power-494-bus	494	29	37	21	67	69	19
67	power-662-bus	662	27	42	19	60	76	19
68	power-685-bus	685	17	21	17	50	61	16
69	power-US-Grid	4941	141	364	43	497	486	39
70	power- bcspwr09	1723	76	50	25	174	178	25
71	power- bcspwr10	5300	100	69	37	375	385	34
72	copresence- InVS13	95	2	2	2	1	1	2
73	hospital-ward- proximity	75	3	3	4	1	1	4
74	infect-hyper	113	4	5	5	1	1	6
75	primary-school- proximity	242	4	5	6	2	1	6
76	road City of Oldenburg	6105	120	177	55	526	630	55
77	road beijing	5036	101	92	41	444	448	45
78	road-chesapeake	39	3	2	3	1	1	3
79	road-euroroad	1039	25	67	25	90	134	22
80	road-minnesota	2640	40	46	32	199	267	31
81	tech-routers-rf	2113	135	278	22	97	176	20
82	email-dnc	1833	18	67	17	32	78	13
83	email-univ	1133	14	49	9	5	72	13

84	inf-USAir97	332	4	27	6	7	20	7
85	inf-euroroad	1039	25	67	22	108	125	22
86	inf-openflights	2905	29	194	25	63	159	27
87	inf-power	4941	141	364	42	504	490	40
88	london_ overground	369	21	45	17	55	56	19
89	london_ underground	271	19	31	16	42	43	17
90	web-EPA	4253	113	206	27	112	207	30
91	web-indochina-2004	11358	1105	996	95	847	929	71
92	web-spam	4767	28	595	24	56	308	32
93	web-webbase-2001	16062	1327	387	81	682	537	71
94	soc-dolphins	62	4	4	5	5	6	5
95	soc-firm-hi-tech	33	3	5	4	2	4	3
96	soc-hamsterster	2000	28	202	28	57	155	22
97	soc-karate	34	4	5	4	2	3	4
98	soc-tribes	16	3	3	3	1	1	3
99	soc-wiki-Vote	889	12	42	9	7	74	10
100	fb-pages-food	620	62	54	18	29	54	18
101	fb-pages-politician	5908	205	260	29	141	241	30
102	fb-pages-public-figure	11565	295	889	42	179	724	34
103	fb-pages-tvshow	3892	508	443	49	238	290	46
104	socfb-Reed98	962	4	78	6	1	19	6
105	socfb-nips-ego	2888	17	6	8	10	11	8
106	rt-retweet	96	10	10	8	14	16	8
107	rt-twitter-copen	761	42	108	21	82	100	20
108	rt_ assad	2139	50	217	30	139	190	28
109	rt_ damascus	3052	399	90	27	149	199	29
110	rt_ obama	3212	254	102	40	192	236	39
111	rt_ occupy	3225	268	277	45	270	309	40
112	rt_ voteonedirection	2280	66	105	32	69	123	32
113	C125-9	125	2	2	3	1	1	3
114	MANN-a9	45	1	1	3	1	1	3
115	brock200-1	200	3	3	5	1	1	5
116	gen200-p0-9-44	200	2	1	4	1	1	4
117	johnson8-4-4	70	5	1	5	1	1	5
118	san200-0-9-1	200	2	2	2	1	1	2
119	bio-celegansneural	297	4	22	6	1	7	6
120	chesapeake	39	3	2	3	1	1	4
121	delaunay_n10	1024	14	14	16	44	58	13
122	delaunay_n11	2048	18	23	18	109	110	19
123	insecta-ant-trophallaxis-colony1	41	3	6	4	1	1	4
124	insecta-ant-trophallaxis-colony2	39	2	6	4	1	1	5
125	mammalia-raccoon-proximity	24	2	2	2	1	1	2

126	reptilia-tortoise- network-bsv	121	9	16	9	5	14	7
127	reptilia-tortoise- network-cs	31	5	5	4	1	2	4
128	reptilia-tortoise- network-fi	496	54	39	23	56	62	21
129	reptilia-tortoise- network-hw	6	2	2	2	1	1	2
130	reptilia-tortoise- network-lm	41	4	7	5	2	7	5
131	reptilia-tortoise- network-pv	22	3	4	3	3	3	4
132	DD199	841	36	41	25	92	87	24
133	DD68	775	24	42	21	65	68	21
134	DD687	725	16	30	16	63	58	16
135	citeseer	2110	165	270	40	197	243	36
136	cora	2485	149	163	26	169	242	30
137	gene	814	67	91	22	85	105	23
138	internet- industry- partnerships	219	6	32	8	2	23	7
139	webkb-wisc	251	13	31	12	10	35	12
140	tech-WHOIS	7476	102	453	24	43	297	19

Leading Eigenvector method, $\rho_p=0.2304$, $\rho_s=0.8497$, QCR=0.9857

Fig. S1. Correlations between the relative path multiplicity and community number estimated by the Leading Eigenvector method in 140 real-world networks. The metadata and results for all 140 networks are provided in the Supplementary Information. Each panel presents a scatter plot on the left, illustrating the correlation between the relative path multiplicity and community number (detected by Leading Eigenvector method) based on the given metric and their RPMI values. The networks are categorized into four quadrants, divided by the median values of community number n_c and the RPMI $\tilde{\Phi}(G)$. The right-hand violin plots show the distribution, mean and median values of community number, which are divided into two parts based on the median value of the RPMI. If there are more overlap between two violin subplots, it suggests a weaker correlation between community number and the RPMI. All community detection packages/functions and parameter settings used in this study, together with brief documentation, are available at <https://github.com/gituserbnu/path-multiplicity.git>.

Walktrap method, $\rho_p=0.0425$, $\rho_s=0.8246$, QCR=0.9143

Fig. S2. Correlations between the relative path multiplicity and community number estimated by the Walktrap method in 140 real-world networks. The metadata and results for all 140 networks are provided in the Supplementary Information. Each panel presents a scatter plot on the left, illustrating the correlation between the relative path multiplicity and community number (detected by Walktrap method) based on the given metric and their RPMI values. The networks are categorized into four quadrants, divided by the median values of community number n_c and the RPMI $\tilde{\Phi}(G)$. The right-hand violin plots show the distribution, mean and median values of community number, which are divided into two parts based on the median value of the RPMI. If there are more overlap between two violin subplots, it suggests a weaker correlation between community number and the RPMI. All community detection packages/functions and parameter settings used in this study, together with brief documentation, are available at <https://github.com/gituserbnu/path-multiplicity.git>.

Fig. S3. Correlations between the relative path multiplicity and community number estimated by the Leiden method in 140 real-world networks. The metadata and results for all 140 networks are provided in the Supplementary Information. Each panel presents a scatter plot on the left, illustrating the correlation between the relative path multiplicity and community number (detected by Leiden method) based on the given metric and their RPMI values. The networks are categorized into four quadrants, divided by the median values of community number n_c and the RPMI $\tilde{\Phi}(G)$. The right-hand violin plots show the distribution, mean and median values of community number, which are divided into two parts based on the median value of the RPMI. If there are more overlap between two violin subplots, it suggests a weaker correlation between community number and the RPMI. All community detection packages/functions and parameter settings used in this study, together with brief documentation, are available at <https://github.com/gituserbnu/path-multiplicity.git>.

Label Propagation method, $\rho_p=0.1913$, $\rho_s=0.8450$, QCR=0.9357

Fig. S4. Correlations between the relative path multiplicity and community number estimated by the Label Propagation method in 140 real-world networks. The metadata and results for all 140 networks are provided in the Supplementary Information. Each panel presents a scatter plot on the left, illustrating the correlation between the relative path multiplicity and community number (detected by Label Propagation method) based on the given metric and their RPMI values. The networks are categorized into four quadrants, divided by the median values of community number n_c and the RPMI $\bar{\Phi}(G)$. The right-hand violin plots show the distribution, mean and median values of community number, which are divided into two parts based on the median value of the RPMI. If there are more overlap between two violin subplots, it suggests a weaker correlation between community number and the RPMI. All community detection packages/functions and parameter settings used in this study, together with brief documentation, are available at <https://github.com/gituserbnu/path-multiplicity.git>.

Infomap method, $\rho_p=0.1578$, $\rho_s=0.8642$, QCR=0.9429

Fig. S5. Correlations between the relative path multiplicity and community number estimated by the Infomap method in 140 real-world networks. The metadata and results for all 140 networks are provided in the Supplementary Information. Each panel presents a scatter plot on the left, illustrating the correlation between the relative path multiplicity and community number (detected by Infomap method) based on the given metric and their RPMI values. The networks are categorized into four quadrants, divided by the median values of community number n_c and the RPMI $\tilde{\Phi}(G)$. The right-hand violin plots show the distribution, mean and median values of community number, which are divided into two parts based on the median value of the RPMI. If there are more overlap between two violin subplots, it suggests a weaker correlation between community number and the RPMI. All community detection packages/functions and parameter settings used in this study, together with brief documentation, are available at <https://github.com/gituserbnu/path-multiplicity.git>.

Louvain method, $\rho_p=0.3012$, $\rho_s=0.8572$, QCR=0.9643

Fig. S6. Correlations between the relative path multiplicity and community number estimated by the Louvain method in 140 real-world networks. The metadata and results for all 140 networks are provided in the Supplementary Information. Each panel presents a scatter plot on the left, illustrating the correlation between the relative path multiplicity and community number (detected by Louvain method) based on the given metric and their RPMI values. The networks are categorized into four quadrants, divided by the median values of community number n_c and the RPMI $\tilde{\Phi}(G)$. The right-hand violin plots show the distribution, mean and median values of community number, which are divided into two parts based on the median value of the RPMI. If there are more overlap between two violin subplots, it suggests a weaker correlation between community number and the RPMI. All community detection packages/functions and parameter settings used in this study, together with brief documentation, are available at <https://github.com/gituserbnu/path-multiplicity.git>.

Comment 3: I asked the authors to justify why QCR was chosen. They responded by “...Its value lies in complementing standard coefficients such as Pearson’s correlation coefficient and Spearman’s correlation coefficient: when associations are monotonic but nonlinear, masked by heavy tails or outliers, or split across clusters that depress the correlation coefficient, QCR can still reveal systematic alignment between variables...” First of all, Spearman’s correlation coefficient already takes nonlinear monotonicity into account, so this argument is not convincing. I am asking for a fundamental reason why QCR is the appropriate metric for this system: why should QCR be high, and what physical or structural insight does that convey? Without a deeper justification, the use of QCR appears superficial and selective.

Response 3: Thank you for your thoughtful comment, which has helped us better explain our choice of the QCR in our study. We acknowledge that Spearman’s correlation captures monotonic nonlinear relationships, and in the revised manuscript we have now included Spearman’s analysis as you suggested. At the same time, we retained QCR because it provides a different and meaningful perspective for our data.

While Spearman’s correlation assesses rank-order agreement across the full range of values, QCR measures how often two variables lie on the same side of their respective medians—that is, whether they tend to be “both high” or “both low” simultaneously. This makes QCR particularly informative when the relationship is driven by coordinated shifts between states, rather than by a smooth trend across all values. In our study, a high QCR indicates that the network metric and PMI frequently move together into similar high or low regimes, reflecting state-like coordination in the system’s behavior—an insight that complements what Spearman’s correlation reveals. Additionally, QCR is inherently robust to outliers and heavy-tailed distributions, as it depends only on whether an observation is above or below the median, not on its exact magnitude or rank. Therefore, QCR was not chosen as a replacement for Spearman’s correlation, but as a complementary tool specifically suited to reveal coordinated state changes—a pattern that conventional correlation measures often miss.

In the revised manuscript, we simultaneously present Pearson’s correlation coefficient (ρ_p), Spearman’s correlation coefficient (ρ_s), and the QCR, and provide a clearer explanation of the distinct information each metric offers. The changes can be found in *Sec. 2.1* and *2.2*. We appreciate your suggestion, which has strengthened the justification of our methodological approach.

Sec. 2.1 Concept of relative path multiplicity

The PMI reflects the complexity of path selection within the network: a large PMI suggests that the network has a hesitant-world property, with many shortest paths between node pairs, potentially leading to a more complex decision-making process. To reveal the relationships

between the PMI and classical network metrics, we first present corresponding scatter plots along with the Pearson's correlation coefficient ρ_p (Pearson et al., *Proc. R. Soc. Lond.* 58, 240–242, 1895), Spearman's correlation coefficient ρ_s (Spearman et al., *Am. J. Psychol.* 15(1), 72–101, 1904), and the quadrant count ratio (QCR) (Holmes et al., *Teach. Stat.* 23, 2001; Choudhary, et al., *Wiley Online Library*, 2017) in 140 real-world networks in Fig. 1a. Pearson's correlation coefficient quantifies the strength of a linear relationship between two variables, whereas Spearman's correlation coefficient is a nonparametric rank-based measure of monotonic association and is often preferred when nonlinearity or outliers make Pearson's coefficient undesirable. The QCR provides a coarse-grained measure of association by dichotomizing each variable at a central location (typically the median) and quantifying how often paired observations fall in the same corresponding half (see the Methods section for details). Together, Pearson's correlation, Spearman's correlation, and QCR offer complementary perspectives on association by targeting linear association, monotonic rank-based association, and median-split concordance, respectively. Consistent results across these measures provide a robustness check for the reported association. Overall, the scatter plots exhibit a disordered distribution, with points broadly dispersed and showing no clear trend. Moreover, we can observe that all three correlation metrics are relatively low, indicating that there is no significant correlation between PMI and classical network metrics.

Fig. 1. Potential factors influencing path multiplicity. a, The scatter plots show the correlation between path multiplicity and classical network metrics in 140 real-world networks. As a reference, the median values of the PMI $\Phi(G)$ and classical network metrics are shown as dashed lines. These scatter plots reveal that the points are widely distributed throughout the plane in all four quadrants, suggesting that there are no significant correlations between path multiplicity and classical network metrics. The metadata and results for all 140 networks are provided in the Supplementary Information. The community detection algorithm used in this study is Newman's deterministic leading eigenvector modularity algorithm (Newman et al., *Phys. Rev. E*, 74, 2006); the specific algorithmic steps and parameter settings are detailed in the Methods section. b, The example simple network is initially a chain network with an initial PMI value of 1. As the edge density increases to a fully connected network, the corresponding PMI value eventually returns to 1. c, The PMI is plotted as a function of the edge density p for three classical network models, ER random networks, regular ring lattices, and CBA scale-free networks, with different network sizes $N = 100, 500, 1000$. For all of the model networks, the PMI changes with the edge density p in some kind of complex non-linear relationship forms.

Sec. 2.2 Intrinsic Factors of Path Multiplicity

To investigate the intrinsic factors influencing path multiplicity in real-world networks, we present scatter plots for the RPMI and classical network metrics along with Pearson's correlation coefficients, Spearman's correlation coefficient, and the QCR for 140 real-world networks in Fig.2 (see the Supplementary Information for the metadata). Given that real-world networks can be disconnected, we focus only on the giant connected component of each network to ensure consistent comparisons. First, it is easy to see that these network metrics are not significantly linearly correlated with path multiplicity, where the highest Pearson's correlation coefficient is $\rho_p = 0.3164$. By inspecting Spearman's correlation coefficients, we find that the community number, global efficiency, average shortest path length, and network diameter are correlated with path multiplicity with $\rho_s > 0.6$. Furthermore, using QCR, we observe that the association between the number of communities and path multiplicity reaches a striking value of 0.9857, where the data points are mostly located in the first and third quadrants in the scatter plot shown in Fig. 2a. However, the majority of the QCR values for the other metrics are below 0.9, suggesting that the number of communities is more significantly correlated with the RPMI. In addition to the scatter plots, we display the violin plots of each network metric, which are divided into two parts on the basis of the median values of the RPMI. If a stronger correlation exists between a network metric and the RPMI, the overlap between two violin subplots should be small. As shown in Fig. 2, compared with other network metrics, the narrow overlap corresponding to the number of communities further demonstrates that the community structure (Rosvall et al., *Proc. Natl. Acad. Sci.* 105, 2008; Girvan et al., *Proc. Natl. Acad. Sci.* 99, 2002) plays a key role in determining path multiplicity.

Fig. 2. Correlations between the relative path multiplicity and classical network metrics in 140 real-world networks. The metadata and results for all 140 networks are provided in the Supplementary Information. The metrics include: a, the community number n_c , b, the average degree $\langle k \rangle$, c, the average shortest path length L , d, the global efficiency E_{glob} , e, the diameter D , f, the assortativity coefficient r , g, the clustering coefficient C , and h, the k-shell index k_s . Each panel (a-h) presents a scatter plot on the left, illustrating the correlation between the relative path multiplicity and the classical network metrics based on the given metric and their RPMI values. The networks are categorized into four quadrants, divided by the median values of the given metric and the RPMI $\tilde{\Phi}(G)$. The right-hand violin plots show the distribution, mean and median values of the corresponding network metric, which are divided into two parts based on the median value of the RPMI. If there are more overlap between two violin subplots, it suggests a weaker correlation between a network metric and the RPMI.

Comment 4: I asked the authors to reconsider the terminology “path hesitation.” The authors insisted on keeping it, but the term remains confusing and has not been properly motivated or defined. For example, the phrase “hesitant-world” abruptly appears on page 3 without any definition regarding what it implies physically or structurally. If the authors wish to retain this terminology, it must be clearly spelled out.

Response 4: We sincerely thank the reviewer for the helpful comment. We agree that the previous terminology “path hesitation” was insufficiently motivated and could lead to unnecessary conceptual ambiguity. In the revised manuscript, we have thoroughly removed this terminology and adopted the standard term “path multiplicity” across all definitions, descriptions, and analyses. Correspondingly, all symbols and abbreviations previously derived from the hesitation terminology have been systematically updated. For example, the quantity formerly denoted as PHA (“path hesitation amount”) is now renamed PMA (“path multiplicity amount”), reflecting its precise mathematical meaning as the multiplicity of shortest paths between node pairs. Likewise, we have renamed PHI (“path hesitation index”) as PMI (“path multiplicity index”) to ensure consistent terminology throughout the manuscript. Overall, we have conducted a manuscript-wide revision to ensure that all notation consistently aligns with the standard terminology. Also, we have modified the second paragraph of the *Introduction*. We are grateful to the reviewer for this comment, which has significantly improved the clarity and precision of the manuscript.

✧ **Reviewer #3:**

Comment 1: I co-reviewed this manuscript with one of the reviewers who provided the listed reports. This is part of the Nature Communications initiative to facilitate training in peer review and to provide appropriate recognition for Early Career Researchers who co-review manuscripts.

Response 1: Thank you for your insightful comments. We have provided a point-by-point response to every comment in the reports and indicated the corresponding revisions with page/line references; all changes are highlighted in the manuscript.

In my opinion, the authors have not yet fully addressed the concerns and comments raised in the first round of review. Therefore, I believe another major revision is necessary. My specific reasons are as follows:

1. In the previous review, we emphasized that introducing a new structural metric, particularly one restricted to unweighted networks, requires strong justification of its utility beyond existing descriptors. --- However, the authors provided only a brief discussion of potential applications without any concrete analysis, limiting the persuasiveness of their argument. It would greatly enhance the manuscript if the authors could demonstrate the metric's practical relevance through at least one specific application scenario.

2. If demonstrating practical application proves challenging, the manuscript should instead clarify and strengthen its theoretical contributions. Currently, these theoretical contributions remain limited and somewhat ambiguous:

2.1 The proposed Relative Path Hesitation Index (RPHI), which normalizes the observed PHI using an equivalent Erdős–Rényi (ER) random graph, *is listed as one of the major contributions to this paper*. However, *employing random-graph null models as baselines for normalizing network metrics* is already *a common and intuitive practice widely documented* in network science literature.

2.2 Another main finding in this paper as claimed by the author "*We find that compared with other metrics, the community structure is more strongly correlated with path multiplicity*" --- which remains largely descriptive and correlation-based. In particular, the authors computed their proposed metric alongside several established metrics across multiple networks and identified correlations. Such an approach, although clear and straightforward, offers limited theoretical depth. Moreover, in the rebuttal, the authors responded by stating, "*deriving an analytical relationship between path multiplicity and community-structure parameters remains a challenging open problem and appears to require additional theoretical tools*," without directly addressing the concern. To significantly strengthen the theoretical grounding, the authors should provide clearer theoretical analysis distinguishing their proposed metric from existing community-based metrics. Without this clarification, *it remains unclear why the proposed metric is preferable to long-established measures directly based on community structure (why not choose classical metric directly?)*. This naturally leads to the third issue as follows:

2.3 The Tribal Scale-Free (TSF) model described in the manuscript appears closely related conceptually to existing network models designed to integrate community structure with realistic, heterogeneous degree distributions. Specifically, it resembles well-known models such as the Degree-Corrected Stochastic Block Model (DC-SBM) and models of Hierarchical Modular Networks (HMN), as well as earlier constructive models like the "Connected Caveman" model. The authors need to explicitly clarify *how the TSF model is theoretically distinct from these classical models*, or

clearly state *whether it is primarily an incremental extension of these existing frameworks*.

2.4 As for the performance of the TSF model that is designed with explicit modularity, outperforms ER, NW, and BA models, which lack strong inherent community structure. *This result is expected with no surprises*. A more rigorous evaluation would involve comparing the TSF model against established models for generating modular networks, such as the Stochastic Block Model (SBM). The authors should discuss how the TSF model compares to SBM variants and whether the specific internal structure of the TSF modules (scale-free via Configuration Model) is necessary for high RPHI, or if modularity alone (as in a standard SBM) is sufficient.

2.5 The authors propose an "interface-driven effect" as the mechanism, where combinations of equivalent path segments across community boundaries multiply the shortest path counts. While plausible, the analysis focuses mainly on the correlation with the number of communities. However, *community structure is more than just how many modules there are*. I'm thinking of modularity, distribution of community sizes, or cut sizes (number of edges crossing communities), which may be the quantities more directly related to the "interface-driven effect".